# THEORETICAL CHARACTERIZATION OF HOW NEURAL NETWORK PRUNING AFFECTS ITS GENERALIZATION

## ABSTRACT

It has been observed in practice that applying pruning-at-initialization methods to neural networks and training the sparsified networks can not only retain the testing performance of the original dense models, but also sometimes even slightly boost the generalization performance. Theoretical understanding for such experimental observations are yet to be developed. This work makes the first attempt to study how different pruning fractions affect the model's gradient descent dynamics and generalization. Specifically, this work considers a classification task for overparameterized two-layer neural networks, where the network is randomly pruned according to different rates at the initialization. It is shown that as long as the pruning fraction is below a certain threshold, gradient descent can drive the training loss toward zero and the network exhibits good generalization performance. More surprisingly, the generalization bound gets better as the pruning fraction gets larger. To complement this positive result, this work further shows a negative result: there exists a large pruning fraction such that while gradient descent is still able to drive the training loss toward zero (by memorizing noise), the generalization performance is no better than random guessing. This further suggests that pruning can change the feature learning process, which leads to the performance drop of the pruned neural network. Up to our knowledge, this is the **first** generalization result for pruned neural networks, suggesting that pruning can improve the neural network's generalization.

## 1 INTRODUCTION

Neural network pruning can be dated back to the early stage of the development of neural networks (LeCun et al., 1989). Since then, many research works have been focusing on using neural network pruning as a model compression technique, e.g. (Molchanov et al., 2019; Luo & Wu, 2017; Ye et al., 2020; Yang et al., 2021). However, all these work focused on pruning neural networks after training to reduce inference time, and, thus, the efficiency gain from pruning cannot be directly transferred to the training phase. It is not until the recent days that Frankle & Carbin (2018) showed a surprising phenomenon: a neural network pruned at the initialization can be trained to achieve competitive performance to the dense model. They called this phenomenon the lottery ticket hypothesis. The lottery ticket hypothesis states that there exists a sparse subnetwork inside a dense network at the random initialization stage such that when trained in isolation, it can match the test accuracy of the original dense network after training for at most the same number of iterations. On the other hand, the algorithm Frankle & Carbin (2018) proposed to find the lottery ticket requires many rounds of pruning and retraining which is computationally expensive. Many subsequent works focused on developing new methods to reduce the cost of finding such a network at the initialization (Lee et al., 2018; Wang et al., 2019; Tanaka et al., 2020; Liu & Zenke, 2020; Chen et al., 2021b). A further investigation by Frankle et al. (2020) showed that some of these methods merely discover the layer-wise pruning ratio instead of sparsity pattern.

The discovery of the lottery ticket hypothesis sparkled further interest in understanding this phenomenon. Another line of research focused on finding a subnetwork inside a dense network at the random initialization such that the subnetwork can achieve good performance (Zhou et al., 2019; Ramanujan et al., 2020). Shortly after that, Malach et al. (2020) formalized this phenomenon which they called the strong lottery ticket hypothesis: under certain assumption on the weight initialization distribution, a sufficiently overparameterized neural network at the initialization contains a subnet-

work with roughly the same accuracy as the target network. Later, Pensia et al. (2020) improved the overparameterization parameters and Sreenivasan et al. (2021) showed that such a type of result holds even if the weight is binary. Unsurprisingly, as it was pointed out by Malach et al. (2020), finding such a subnetwork is computationally hard. Nonetheless, all of the analysis is from a function approximation perspective and none of the aforementioned works have considered the effect of pruning on gradient descent dynamics, let alone the neural networks' generalization.

Interestingly, via empirical experiments, people have found that sparsity can further improve generalization in certain scenarios (Chen et al., 2021a; Ding et al., 2021; He et al., 2022). There have also been empirical works showing that *random pruning* can be effective (Frankle et al., 2020; Su et al., 2020; Liu et al., 2021b). However, theoretical understanding of such benefit of pruning of neural networks is still limited. In this work, we take the first step to answer the following important open question from a theoretical perspective:

> *How does pruning fraction affect the training dynamics and the model's generalization, if the model is pruned at the initialization and trained by gradient descent?*

We study this question using random pruning. We consider a classification task where the input data consists of class-dependent sparse signal and random noise. We analyze the training dynamics of a two-layer convolutional neural network pruned at the initialization. Specifically, this work makes the following contributions:

- **Mild pruning.** We prove that there indeed exists a range of pruning fraction where the pruning fraction is small and the generalization error bound gets better as pruning fraction gets larger. In this case, the signal in the feature is well-preserved and due to the effect of pruning purifying the feature, the effect from noise is reduced. We provide detailed explanation in Section 3. Up to our knowledge, this is the first theoretical result on generalization for *pruned* neural networks, which suggests that pruning can improve generalization under some setting. Further, we conduct experiments to verify our results.

- **Over pruning.** To complement the above positive result, we also show a negative result: if the pruning fraction is larger than a certain threshold, then the generalization performance is no better than a simple random guessing, although gradient descent is still able to drive the training loss toward zero. This further suggests that the performance drop of the pruned neural network is not solely caused by the pruned network's own lack of trainability or expressiveness, but also by the change of gradient descent dynamics due to pruning.

- Technically, we develop novel analysis to bound pruning effect to weight-noise and weight-signal correlation. Further, in contrast to many previous works that considered only the binary case, our analysis handles multi-class classification with general cross-entropy loss. Here, a key technical development is a gradient upper bound for multi-class cross-entropy loss, which might be of independent interest.

Pictorially, our result is summarized in Figure 1. We point out that the neural network training we consider is in the *feature learning* regime, where the weight parameters can go far away from their initialization. This is *fundamentally different* from the popular *neural tangent kernel* regime, where the neural networks essentially behave similar to its linearization.

## 1.1 RELATED WORKS

**The Lottery Ticket Hypothesis and Sparse Training.** The discovery of the lottery ticket hypothesis (Frankle & Carbin, 2018) has inspired further investigation and applications. One line of research has focused on developing computationally efficient methods to enable sparse training: the static sparse training methods are aiming at identifying a sparse mask at the initialization stage based on different criterion such as SNIP (loss-based) (Lee et al., 2018), GraSP (gradient-based) (Wang et al., 2019), SynFlow (synaptic strength-based) (Tanaka et al., 2020), neural tangent kernel based method (Liu & Zenke, 2020) and one-shot pruning (Chen et al., 2021b). Random pruning has also been considered in static sparse training such as uniform pruning (Mariet & Sra, 2015; He et al., 2017; Gale et al., 2019; Suau et al., 2018), non-uniform pruning (Mocanu et al., 2016), expander-graph-related techniques (Prabhu et al., 2018; Kepner & Robinett, 2019) Erdös-Rényi (Mocanu et al., 2018) and Erdös-Rényi-Kernel (Evci et al., 2020). On the other hand, dynamic sparse training allows the

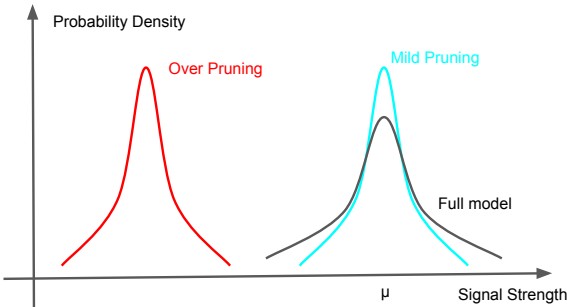

Figure 1: A pictorial demonstration of our results. The bell-shaped curves model the distribution of the signal in the features, where the mean represents the signal strength and the width of the curve indicates the variance of noise. Our results show that mild pruning preserves the signal strength and reduces the noise variance (and hence yields better generalization), whereas over pruning lowers signal strength albeit reducing noise variance.

sparse mask to be updated (Mocanu et al., 2018; Mostafa & Wang, 2019; Evci et al., 2020; Jayakumar et al., 2020; Liu et al., 2021c;d;a; Peste et al., 2021). The sparsity pattern can also be learned by using sparsity-inducing regularizer (Yang et al., 2020). Recently, He et al. (2022) discovered that pruning can exhibit a double descent phenomenon when the data-set labels are corrupted.

Another line of research has focused on studying pruning the neural networks at its random initialization to achieve good performance (Zhou et al., 2019; Ramanujan et al., 2020). In particular, Ramanujan et al. (2020) showed that it is possible to prune a randomly initialized wide ResNet-50 to match the performance of a ResNet-34 trained on ImageNet. This phenomenon is named the strong lottery ticket hypothesis. Later, Malach et al. (2020) proved that under certain assumption on the initialization distribution, a target network of width $d$ and depth $l$ can be approximated by pruning a randomly initialized network that is of a polynomial factor (in $d, l$) wider and twice deeper even without any further training. However finding such a network is computationally hard, which can be shown by reducing the pruning problem to optimizing a neural network. Later, Pensia et al. (2020) improved the widening factor to being logarithmic and Sreenivasan et al. (2021) proved that with a polylogarithmic widening factor, such a result holds even if the network weight is binary. A follow-up work shows that it is possible to find a subnetwork achieving good performance at the initialization and then fine-tune (Sreenivasan et al., 2022). Our work, on the other hand, analyzes the gradient descent dynamics of a pruned neural network and its generalization after training.

**Analyses of Training Neural Networks by Gradient Descent.** A series of work (Allen-Zhu et al., 2019; Du et al., 2019; Lee et al., 2019; Zou et al., 2020; Zou & Gu, 2019; Ji & Telgarsky, 2019; Chen et al., 2020b; Song & Yang, 2019; Oymak & Soltanolkotabi, 2020) has proved that if a deep neural network is wide enough, then (stochastic) gradient descent provably can drive the training loss toward zero in a fast rate based on neural tangent kernel (NTK) (Jacot et al., 2018). Further, under certain assumption on the data, the learned network is able to generalize (Cao & Gu, 2019; Arora et al., 2019). However, as it is pointed out by Chizat et al. (2019), in the NTK regime, the gradient descent dynamics of the neural network essentially behaves similarly to its linearization and the learned weight is not far away from the initialization, which prohibits the network from performing any useful feature learning. In order to go beyond NTK regime, one line of research has focused on the mean field limit (Song et al., 2018; Chizat & Bach, 2018; Rotskoff & Vanden-Eijnden, 2018; Wei et al., 2019; Chen et al., 2020a; Sirignano & Spiliopoulos, 2020; Fang et al., 2021). Recently, people have started to study the neural network training dynamics in the feature learning regime where data from different class is defined by a set of class-related signals which are low rank (Allen-Zhu & Li, 2020; 2022; Cao et al., 2022; Shi et al., 2021; Telgarsky, 2022). However, all previous works did not consider the effect of pruning. Our work also focuses on the aforementioned feature learning regime, but for the first time characterizes the impact of pruning on the generalization performance of neural networks.

## 2 PRELIMINARIES AND PROBLEM FORMULATION

In this section, we introduce our notation, data generation process, neural network architecture and the optimization algorithm.

**Notations.** We use lower case letters to denote scalars and boldface letters and symbols (e.g. $\mathbf{x}$) to denote vectors and matrices. We use $\odot$ to denote element-wise product. For an integer $n$, we use $[n]$ to denote the set of integers $\{1, 2, \ldots, n\}$. We use $x = O(y), x = \Omega(y), x = \Theta(y)$ to denote that there exists a constant $C$ such that $x \leq Cy$, $x \geq Cy, x = Cy$ respectively. We use $\widetilde{O}, \widetilde{\Omega}$ and $\widetilde{\Theta}$ to hide polylogarithmic factor in these notations. Finally, we use $x = \text{poly}(y)$ if $x = O(y^C)$ for some positive constant $C$, and $x = \text{poly} \log y$ if $x = \text{poly}(\log y)$.

## 2.1 SETTINGS

**Definition 2.1** (Data distribution of $K$ classes). *Consider we are given the set of signal vectors $\{\mu \mathbf{e}_i\}_{i=1}^K$, where $\mu > 0$ denotes the strength of the signal, and $\mathbf{e}_i$ denotes the $i$-th standard basis vector with its $i$-th entry being 1 and all other coordinates being 0. Each data point $(\mathbf{x}, y)$ with $\mathbf{x} = [\mathbf{x}_1^\top, \mathbf{x}_2^\top]^\top \in \mathbb{R}^{2d}$ and $y \in [K]$ is generated from the following distribution $\mathcal{D}$:*

1. *The label $y$ is generated from a uniform distribution over $[K]$.*

2. *A noise vector $\boldsymbol{\xi}$ is generated from the Gaussian distribution $\mathcal{N}(\mathbf{0}, \sigma_n^2 \mathbf{I})$.*

3. *With probability $1/2$, assign $\mathbf{x}_1 = \boldsymbol{\mu}_y$, $\mathbf{x}_2 = \boldsymbol{\xi}$; with probability $1/2$, assign $\mathbf{x}_2 = \boldsymbol{\mu}_y$, $\mathbf{x}_1 = \boldsymbol{\xi}$ where $\boldsymbol{\mu}_y = \mu \mathbf{e}_y$.*

The sparse signal model is motivated by the empirical observation that during the process of training neural networks, the output of each layer of ReLU is usually sparse instead of dense. This is partially due to the fact that in practice the bias term in the linear layer is used (Song et al., 2021). For samples from different classes, usually a different set of neurons fire. Our study can be seen as a formal analysis on pruning the second last layer of a deep neural network in the layer-peeled model as in Zhu et al. (2021); Zhou et al. (2022). We also point out that our assumption on the sparsity of the signal is necessary for our analysis. If we don't have this sparsity assumption and only make assumption on the $\ell_2$ norm of the signal, then in the extreme case, the signal is uniformly distributed across all coordinate and the effect of pruning to the signal and the noise will be essentially the same: their $\ell_2$ norm will both be reduced by a factor of $\sqrt{p}$.

**Network architecture and random pruning.** We consider a two-layer convolutional neural network model with polynomial ReLU activation $\sigma(z) = (\max\{0, z\})^q$, where we focus on the case when $q = 3$ [1] The network is pruned at the initialization by mask $\mathbf{M}$ where each entry in the mask $\mathbf{M}$ is generated i.i.d. from Bernoulli$(p)$. Let $\mathbf{m}_{j,r}$ denotes the $r$-th row of $\mathbf{M}_j$. Given the data $(\mathbf{x}, y)$, the output of the neural network can be written as $F(\mathbf{W} \odot \mathbf{M}, \mathbf{x}) = (F_1(\mathbf{W}_1 \odot \mathbf{M}_1, \mathbf{x}), F_2(\mathbf{W}_2 \odot \mathbf{M}_2, \mathbf{x}), \ldots, F_k(\mathbf{W}_k \odot \mathbf{M}_k, \mathbf{x}))$ where the $j$-th output is given by

$$F_j(\mathbf{W}_j \odot \mathbf{M}_j, \mathbf{x}) = \sum_{r=1}^m [\sigma(\langle \mathbf{w}_{j,r} \odot \mathbf{m}_{j,r}, \mathbf{x}_1 \rangle) + \sigma(\langle \mathbf{w}_{j,r} \odot \mathbf{m}_{j,r}, \mathbf{x}_2 \rangle)]$$

$$= \sum_{r=1}^m [\sigma(\langle \mathbf{w}_{j,r} \odot \mathbf{m}_{j,r}, \boldsymbol{\mu} \rangle) + \sigma(\langle \mathbf{w}_{j,r} \odot \mathbf{m}_{j,r}, \boldsymbol{\xi} \rangle)].$$

The mask $\mathbf{M}$ is only sampled once at the initialization and remains fixed through the entire training process. From now on, **we use tilde over a symbol to denote its masked version, e.g., $\widetilde{\mathbf{W}} = \mathbf{W} \odot \mathbf{M}$ and $\widetilde{\mathbf{w}}_{j,r} = \mathbf{w}_{j,r} \odot \mathbf{m}_{j,r}$.**

Since $\boldsymbol{\mu}_j \odot \mathbf{m}_{j,r} = \mathbf{0}$ with probability $1 - p$, some neurons will not receive the corresponding signal at all and will only learn noise. Therefore, for each class $j \in [k]$, we split the neurons into two sets based on whether it receives its corresponding signal or not:

$$\mathcal{S}_{\text{signal}}^j = \{r \in [m] : \boldsymbol{\mu}_j \odot \mathbf{m}_{j,r} \neq \mathbf{0}\}, \qquad \mathcal{S}_{\text{noise}}^j = \{r \in [m] : \boldsymbol{\mu}_j \odot \mathbf{m}_{j,r} = \mathbf{0}\}.$$

**Gradient descent algorithm.** We consider the network is trained by cross-entropy loss with softmax. We denote by $\text{logit}_i(F, \mathbf{x}) := \frac{e^{F_i(\mathbf{x})}}{\sum_{j \in [k]} e^{F_j(\mathbf{x})}}$ and the cross-entropy loss can be written as

---

[1]We point out that as many previous works (Allen-Zhu & Li, 2020; Zou et al., 2021; Cao et al., 2022), polynomial ReLU activation can help us simplify the analysis of gradient descent, because polynomial ReLU activation can give a much larger separation of signal and noise (thus, cleaner analysis) than ReLU. Our analysis can be generalized to ReLU activation by using the arguments in (Allen-Zhu & Li, 2022).

$\ell(F(\mathbf{x}, y)) = -\log \text{logit}_y(F, \mathbf{x})$. The convolutional neural network is trained by minimizing the **empirical cross-entropy loss** given by

$$L_S(\mathbf{W}) = \frac{1}{n} \sum_{i=1}^{n} \ell[F(\mathbf{W} \odot \mathbf{M}; \mathbf{x}_i, y_i)] = \mathbb{E}_S \ell[F(\mathbf{W} \odot \mathbf{M}; \mathbf{x}_i, y_i)],$$

where $S = \{(\mathbf{x}_i, y_i)\}_{i=1}^{n}$ is the training data set. Similarly, we define the **generalization loss** as

$$L_{\mathcal{D}} := \mathbb{E}_{(\mathbf{x}, y)}[\ell(F(\mathbf{W} \odot \mathbf{M}; \mathbf{x}, y))].$$

The model weights are initialized from a i.i.d. Gaussian $\mathcal{N}(0, \sigma_0^2)$. The gradient of the cross-entropy loss is given by $\ell'_{j,i} := \ell'_j(\mathbf{x}_i, y_i) = \text{logit}_j(F, \mathbf{x}_i) - \mathbb{I}(j = y_i)$. Since

$$\nabla_{\mathbf{w}_{j,r}} L_S(\mathbf{W} \odot \mathbf{M}) = \nabla_{\mathbf{w}_{j,r} \odot \mathbf{m}_{j,r}} L_S(\mathbf{W} \odot \mathbf{M}) \odot \mathbf{m}_{j,r} = \nabla_{\widetilde{\mathbf{w}}_{j,r}} L_S(\widetilde{\mathbf{W}}) \odot \mathbf{m}_{j,r},$$

we can write the full-batch gradient descent update of the weights as

$$\begin{aligned}
\widetilde{\mathbf{w}}_{j,r}^{(t+1)} &= \widetilde{\mathbf{w}}_{j,r}^{(t)} - \eta \nabla_{\widetilde{\mathbf{w}}_{j,r}} L_S(\widetilde{\mathbf{W}}) \odot \mathbf{m}_{j,r} \\
&= \widetilde{\mathbf{w}}_{j,r}^{(t)} - \frac{\eta}{n} \sum_{i=1}^{n} \ell'^{(t)}_{j,i} \cdot \sigma'\left(\left\langle \widetilde{\mathbf{w}}_{j,r}^{(t)}, \boldsymbol{\xi}_i \right\rangle\right) \cdot \widetilde{\boldsymbol{\xi}}_{j,r,i} - \frac{\eta}{n} \sum_{i=1}^{n} \ell'^{(t)}_{j,i} \sigma'\left(\left\langle \widetilde{\mathbf{w}}_{j,r}^{(t)}, \boldsymbol{\mu}_{y_i} \right\rangle\right) \boldsymbol{\mu}_{y_i} \odot \mathbf{m}_{j,r},
\end{aligned}$$

for $j \in [K]$ and $r \in [m]$, where $\widetilde{\boldsymbol{\xi}}_{j,r,i} = \boldsymbol{\xi}_i \odot \mathbf{m}_{j,r}$.

**Condition 2.2.** *We consider the parameter regime described as follows: (1) Number of classes $K = O(\log d)$. (2) Total number of training samples $n = \text{poly} \log d$. (3) Dimension $d \geq C_d$ for some sufficiently large constant $C_d$. (4) Relationship between signal strength and noise strength: $\mu = \Theta(\sigma_n \sqrt{d} \log d) = \Theta(1)$. (5) The number of neurons in the network $m = \Omega(\text{poly} \log d)$. (6) Initialization variance: $\sigma_0 = \widetilde{\Theta}(m^{-4} n^{-1} \mu^{-1})$. (7) Learning rate: $\Omega(1/\text{poly}(d)) \leq \eta \leq \widetilde{O}(1/\mu^2)$. (8) Target training loss: $\epsilon = \Theta(1/\text{poly}(d))$.*

Conditions (1) and (2) ensure that there are enough samples in each class with high probability. Condition (3) ensures that our setting is in high-dimensional regime. Condition (4) ensures that the full model can be trained to exhibit good generalization. Condition (5), (6) and (7) ensures that the neural network is sufficiently overparameterized and can be optimized efficiently by gradient descent. Condition (7) and (8) further ensures that training time is polynomial in $d$. We further discuss the practical consideration of $\eta$ and $\epsilon$ to justify their condition in Remark D.9.

## 3 MILD PRUNING

### 3.1 MAIN RESULT

The first main result shows that there exists a threshold on the pruning fraction $p$ such that pruning helps the neural network's generalization.

**Theorem 3.1** (Main Theorem for Mild Pruning, Informal). *Under Condition 2.2, if $p \in [C_1 \frac{\log d}{m}, 1]$ for some constant $C_1$, then with probability at least $1 - O(d^{-1})$ over the randomness in the data, network initialization and pruning, there exists $T = \widetilde{O}(K \eta^{-1} \sigma_0^{2-q} \mu^{-q} + K^2 m^4 \mu^{-2} \eta^{-1} \epsilon^{-1})$ such that*

1. *The training loss is below $\epsilon$: $L_S(\widetilde{\mathbf{W}}^{(T)}) \leq \epsilon$.*

2. *The generalization loss can be bounded by $L_{\mathcal{D}}(\widetilde{\mathbf{W}}^{(T)}) \leq O(K\epsilon) + \exp(-n^2/p)$.*

Theorem 3.1 indicates that there exists a threshold in the order of $\Theta(\frac{\log d}{m})$ such that if $p$ is above this threshold (i.e., the fraction of the pruned weights is small), gradient descent is able to drive the training loss towards zero (as item 1 claims) and the overparameterized network achieves good testing performance (as item 2 claims). In the next subsection, we explain why pruning can help generalization via an outline of our proof, and we defer all the detailed proofs in Appendix D.

### 3.2 PROOF OUTLINE

Our proof contains the establishment of the following two properties:

- First we show that after mild pruning the network is still able to learn the signal, and the magnitude of the signal in the feature is preserved.
- Then we show that given a new sample, pruning reduces the noise effect in the feature which leads to the improvement of generalization.

We first show the above properties for three stages of gradient descent: initialization, feature growing phase, and converging phase, and then establish the generalization property.

**Initialization.** First of all, readers might wonder why pruning can even preserve signal at all. Intuitively, a network will achieve good performance if its weights are highly correlated with the signal (i.e., their inner product is large). Two intuitive but misleading heuristics are given by the following:

- Consider a fixed neuron weight. At the random initialization, in expectation, the signal correlation with the weights is given by $\mathbb{E}_{\mathbf{w},\mathbf{m}}[|\langle \mathbf{w} \odot \mathbf{m}, \boldsymbol{\mu} \rangle|] \leq p\sigma_0\mu$ and the noise correlation with the weights is given by $\mathbb{E}_{\mathbf{w},\mathbf{m},\boldsymbol{\xi}}[|\langle \mathbf{w} \odot \mathbf{m}, \boldsymbol{\xi} \rangle|] \leq \sqrt{\mathbb{E}_{\mathbf{w},\mathbf{m},\boldsymbol{\xi}}[\langle \mathbf{w} \odot \mathbf{m}, \boldsymbol{\xi} \rangle^2]} = \sigma_0\sigma_n\sqrt{pd}$ by Jensen's inequality. Based on this argument, taking a sum over all the neurons, pruning will hurt weight-signal correlation more than weight-noise correlation.
- Since we are pruning with Bernoulli($p$), a given neuron will not receive signal at all with probability $1 - p$. Thus, there is roughly $p$ fraction of the neurons receiving the signal and the rest $1 - p$ fraction will be purely learning from noise. Even though for every neuron, roughly $\sqrt{p}$ portion of $\ell_2$ mass from the noise is reduced, at the same time, pruning also creates $1 - p$ fraction of neurons which do not receive signals at all and will purely output noise after training. Summing up the contributions from every neuron, the signal strength is reduced by a factor of $p$ while the noise strength is reduced by a factor of $\sqrt{p}$. We again reach the conclusion of pruning under any rate will hurt the signal more than noise.

The above analysis shows that under any pruning rate, it seems pruning can only hurt the signal more than noise at the initialization. Such analysis would be indicative if the network training is under the *neural tangent kernel regime*, where the weight of each neuron does not travel far from its initialization so that the above analysis can still hold approximately after training. However, when the neural network training is in the *feature learning regime*, this average type analysis becomes misleading. Namely, in such a regime, the weights with large correlation with the signal at the initialization will quickly evolve into singleton neurons and those weights with small correlation will remain small. In our proof, we focus on the *featuring learning regime*, and analyze how the network weights change and what are the effect of pruning during various stages of gradient descent.

We now analyze the effect of pruning on weight-signal correlation and weight-noise correlation at the initialization. Our first lemma leverages the sparsity of our signal and shows that if the pruning is mild, then it will not hurt the maximum weight-signal correlation much at the initialization. On the other hand, the maximum weight-noise correlation is reduced by a factor of $\sqrt{p}$.

**Lemma 3.2** (Initialization). *With probability at least $1 - 2/d$, for all $i \in [n]$,*

$$\sigma_0\sigma_n\sqrt{pd} \leq \max_r \left\langle \widetilde{\mathbf{w}}_{j,r}^{(0)}, \boldsymbol{\xi}_i \right\rangle \leq \sqrt{2\log(Kmd)}\sigma_0\sigma_n\sqrt{pd}.$$

*Further, suppose $pm \geq \Omega(\log(Kd))$, with probability $1 - 2/d$, for all $j \in [K]$,*

$$\sigma_0 \|\boldsymbol{\mu}_j\|_2 \leq \max_{r \in \mathcal{S}_{\mathrm{signal}}^j} \left\langle \widetilde{\mathbf{w}}_{j,r}^{(0)}, \boldsymbol{\mu}_j \right\rangle \leq \sqrt{2\log(8pmKd)}\sigma_0 \|\boldsymbol{\mu}_j\|_2.$$

Given this lemma, we now prove that there exists at least one neuron that is heavily aligned with the signal after training. Similarly to previous works (Allen-Zhu & Li, 2020; Zou et al., 2021; Cao et al., 2022), the analysis is divided into two phases: feature growing phase and converging phase.

**Feature Growing Phase.** In this phase, the gradient of the cross-entropy is large and the weight-signal correlation grows much more quickly than weight-noise correlation thanks to the polynomial ReLU. We show that the signal strength is relatively unaffected by pruning while the noise level is reduced by a factor of $\sqrt{p}$.

**Lemma 3.3** (Feature Growing Phase, Informal)**.** *Under Condition 2.2, there exists time $T_1$ such that*

1. *The max weight-signal correlation is large:* $\max_r \left\langle \widetilde{\mathbf{w}}_{j,r}^{(T_1)}, \boldsymbol{\mu}_j \right\rangle \geq m^{-1/q}$ *for $j \in [K]$.*

2. *The weight-noise and cross-class weight-signal correlations are small: if $j \neq y_i$, then* $\max_{j,r,i} \left| \left\langle \widetilde{\mathbf{w}}_{j,r}^{(T_1)}, \boldsymbol{\xi}_i \right\rangle \right| \leq O(\sigma_0 \sigma_n \sqrt{pd})$ *and* $\max_{j,r,k} \left| \left\langle \widetilde{\mathbf{w}}_{j,r}^{(T_1)}, \boldsymbol{\mu}_k \right\rangle \right| \leq \widetilde{O}(\sigma_0 \mu)$.

**Converging Phase.** We show that gradient descent can drive the training loss toward zero while the signal in the feature is still large. An important intermediate step in our argument is the development of the following gradient upper bound for multi-class cross-entropy loss which introduces an extra factor of $K$ in the gradient upper bound.

**Lemma 3.4** (Gradient Upper Bound, Informal)**.** *Under Condition 2.2, we have*

$$\left\| \nabla L_S(\widetilde{\mathbf{W}}^{(t)}) \odot \mathbf{M} \right\|_F^2 \leq O(K m^{2/q} \mu^2) L_S(\widetilde{\mathbf{W}}^{(t)}).$$

*Proof Sketch.* To prove this upper bound, note that for a given input $(\mathbf{x}_i, y_i)$, $\ell_{y_i,i}'^{(t)} \nabla F_{y_i}(\mathbf{x}_i)$ should make major contribution to $\left\| \nabla \ell(\widetilde{\mathbf{W}}; \mathbf{x}_i, y_i) \right\|_F$. Further note that $|\ell_{y_i,i}'^{(t)}| = 1 - \text{logit}_{y_i}(F; \mathbf{x}_i) = \frac{\sum_{j \neq y_i} e^{F_j(\mathbf{x}_i)}}{\sum_j e^{F_j(\mathbf{x}_i)}} \leq \frac{\sum_{j \neq y_i} e^{F_j(\mathbf{x}_i)}}{e^{F_{y_i}(\mathbf{x}_i)}}$. Now, apply the property that $F_j(\mathbf{x}_i)$ is small for $j \neq y_i$ (which we prove in the appendix), the numerator will contribute a factor of $K$. To bound the rest, we utilize the special property of multi-class cross-entropy loss: $|\ell_{j,i}'^{(t)}| \leq |\ell_{y_i,i}'^{(t)}| \leq \ell_i^{(t)}$. However, a naive application of this inequality will result in a factor of $K^3$ instead $K$ in our bound. The trick is to further use the fact that $\sum_{j \neq y_i} |\ell_{j,i}'^{(t)}| = |\ell_{y_i,i}'^{(t)}|$. $\square$

Using the above gradient upper bound, we can show that the objective can be minimized.

**Lemma 3.5** (Converging Phase, Informal)**.** *Under Condition 2.2, there exists $T_2$ such that for some time $t \in [T_1, T_2]$ we have*

1. *The results from the feature growing phase (Lemma 3.3) hold up to constant factors.*
2. *The training loss is small $L_S(\widetilde{\mathbf{W}}^{(t)}) \leq \epsilon$.*

Notice that the weight-noise correlation still remains reduced by a factor of $\sqrt{p}$ after training. Lemma 3.5 proves the statement of the training loss in Theorem 3.1.

**Generalization Analysis.** Finally, we show that pruning can purify the feature by reducing the variance of the noise by a factor of $p$ when a new sample is given. The lemma below shows that the variance of weight-noise correlation for the trained weights is reduced by a factor of $p$.

**Lemma 3.6.** *The neural network weight $\widetilde{\mathbf{W}}^\star$ after training satisfies that*

$$\mathbb{P}_{\boldsymbol{\xi}} \left[ \max_{j,r} \left| \langle \widetilde{\mathbf{w}}_{j,r}^\star, \boldsymbol{\xi} \rangle \right| \geq (2m)^{-2/q} \right] \leq 2Km \exp\left( -\frac{(2m)^{-4/q}}{O(\sigma_0^2 \sigma_n^2 pd)} \right).$$

Using this lemma, we can show that pruning yields better generalization bound (i.e., the bound on the generalization loss) claimed in Theorem 3.1.

## 4 Over Pruning

Our second result shows that there exists a relatively large pruning fraction (i.e., small $p$) such that the learned model yields poor generalization, although gradient descent is still able to drive the training error toward zero. The full proof is defered to Appendix E.

**Theorem 4.1** (Main Theorem for Over Pruning, Informal)**.** *Under Condition 2.2 if $p = \Theta(\frac{1}{Km \log d})$, then with probability at least $1 - 1/\text{poly} \log d$ over the randomness in the data, network initialization and pruning, there exists $T = O(\eta^{-1} n \sigma_0^{q-2} \sigma_n^{-q} (pd)^{-q/2} + \eta^{-1} \epsilon^{-1} m^4 n \sigma_n^{-2} (pd)^{-1})$ such that*

1. *The training loss is below $\epsilon$: $L_S(\widetilde{\mathbf{W}}^{(T)}) \leq \epsilon$.*
2. *The generalization loss is large: $L_{\mathcal{D}}(\widetilde{\mathbf{W}}^{(T)}) \geq \Omega(\log K)$.*

**Remark 4.2.** The above theorem indicates that in the over-pruning case, the training loss can still go to zero. However, the generalization loss of our neural network behaves no much better than random guessing, because given any sample, random guessing will assign each class with probability $1/K$, which yields a generalization loss of $\log K$. The readers might wonder why the condition for this to happen is $p = \Theta(\frac{1}{Km\log d})$ instead of $O(\frac{1}{Km\log d})$. Indeed, the generalization will still be bad if $p$ is too small. However, now the neural network is not only unable to learn the signal but also cannot efficiently memorize the noise via gradient descent.

***Proof Outline****.* Now we analyze the over-pruning case. We first show that there is a good chance that the model will not receive any signal after pruning due to the sparse signal assumption and mild overparameterization of the neural network. Then, leveraging such a property, we bound the weight-signal and weight-noise properties for the feature growing and converging phases of gradient descent, as stated in the following two lemmas, respectively. Our result indicates that the training loss can still be driven toward zero by letting the neural network memorize the noise, the proof of which further exploits the fact that high dimensional Gaussian noise are nearly orthogonal.

**Lemma 4.3** (Feature Growing Phase, Informal)**.** *Under Condition 2.2, there exists $T_1$ such that*

- *Some weights has large correlation with noise: $\max_r \left\langle \widetilde{\mathbf{w}}_{y_i,r}^{(T_1)}, \boldsymbol{\xi}_i \right\rangle \geq m^{-1/q}$ for all $i \in [n]$.*
- *The cross-class weight-noise and weight-signal correlations are small: if $j \neq y_i$, then $\max_{j,r,i} \left| \left\langle \widetilde{\mathbf{w}}_{j,r}^{(T_1)}, \boldsymbol{\xi}_i \right\rangle \right| = \widetilde{O}(\sigma_0 \sigma_n \sqrt{pd})$ and $\max_{j,r,k} \left| \left\langle \widetilde{\mathbf{w}}_{j,r}^{(T_1)}, \boldsymbol{\mu}_k \right\rangle \right| \leq \widetilde{O}(\sigma_0 \mu)$.*

**Lemma 4.4** (Converging Phase, Informal)**.** *Under Condition 2.2, there exists a time $T_2$ such that $\exists t \in [T_1, T_2]$, the results from phase 1 still holds (up to constant factors) and $L_S(\widetilde{\mathbf{W}}^{(t)}) \leq \epsilon$.*

Finally, since the above lemmas show that the network is purely memorizing the noise, we further show that such a network yields poor generalization performance as stated in Theorem 4.1. $\qquad\square$

## 5 EXPERIMENTS

### 5.1 SIMULATIONS TO VERIFY OUR RESULTS

In this section, we conduct simulations to verify our results. We conduct our experiment using binary classification task and show that our result holds for ReLU networks. Our experiment settings are the follows: we choose input to be $\mathbf{x} = [\mathbf{x}_1, \mathbf{x}_2] = [y\mathbf{e}_1, \boldsymbol{\xi}] \in \mathbb{R}^{800}$ and $\mathbf{x}_1, \mathbf{x}_2 \in \mathbb{R}^{400}$, where $\boldsymbol{\xi}_i$ is sampled from a Gaussian distribution. The class labels $y$ are $\{\pm 1\}$. We use 100 training examples and 100 testing examples. The network has width 150 and is initialized with random Gaussian distribution with variance 0.01. Then, $p$ fraction of the weights are randomly pruned. We use the learning rate of 0.001 and train the network over 1000 iterations by gradient descent.

The observations are summarized as follows. In Figure 2a, when the noise level is $\sigma_n = 0.5$, the pruned network usually can perform at the similar level with the full model when $p \leq 0.5$ and noticably better when $p = 0.3$. When $p > 0.5$, the test error increases dramatically while the training accuracy still remains perfect. On the other hand, when the noise level becomes large $\sigma_n = 1$ (Figure 2b), the full model can no longer achieve good testing performance but mild pruning can improve the model's generalization. Note that the training accuracy in this case is still perfect (omitted in the figure). We observe that in both settings when the model test error is large, the variance is also large. However, in Figure 2b, despite the large variance, the mean curve is already smooth. In particular, Figure 2c plots the testing error over the training iterations under $p = 0.5$ pruning rate. This suggests that pruning can be beneficial even when the input noise is large.

### 5.2 ON THE REAL WORLD DATASET

To further demonstrate the mild/over pruning phenomenon, we conduct experiments on MNIST (Deng, 2012) and CIFAR-10 (Krizhevsky et al., 2009) datasets. We consider neural network ar-

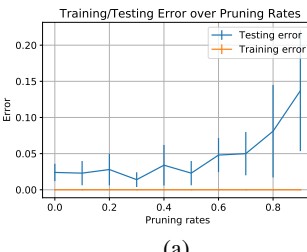 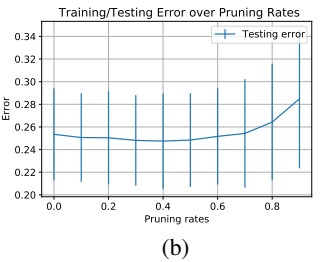 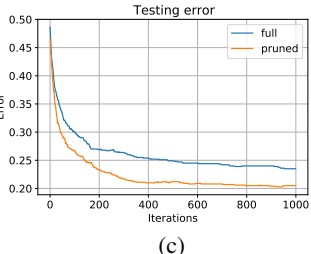

(a)          (b)          (c)

Figure 2: Figure (a) shows the relationship between pruning rates $p$ and training/testing error under noise variance $\sigma_n = 0.5$. Figure (b) shows the relationship between pruning rates $p$ and testing error under noise variance $\sigma_n = 1$. The training error is omitted since it stays effectively at zero across all pruning rates. Figure (c) shows a particular training curve under pruning rate $p = 50\%$ and noise variance $\sigma_n = 1$. Each data point is created by taking an average over 10 independent runs.

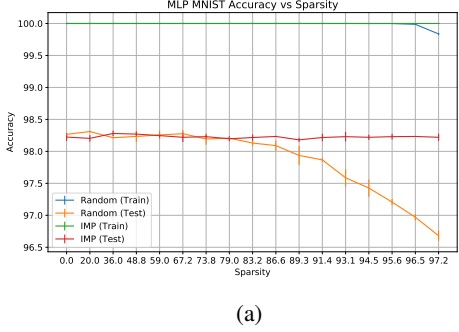          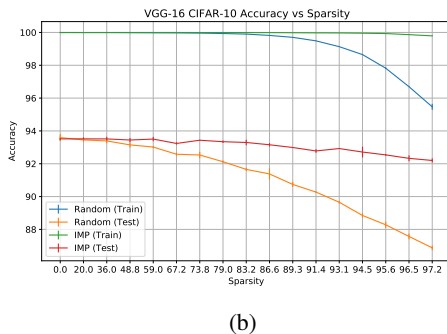

(a)                              (b)

Figure 3: Figure (a) shows the result between pruning rates $p$ and accuracy on MLP-1024-1024 on MNIST. Figure (b) shows the result on VGG-16 on CIFAR-10. Each data point is created by taking an average over 3 independent runs.

chitectures including MLP with 2 hidden layers of width 1024, VGG, ResNets (He et al., 2016) and wide ResNet (Zagoruyko & Komodakis, 2016). In addition to random pruning, we also add iterative-magnitude-based pruning Frankle & Carbin (2018) into our experiments. Both pruning methods are prune-at-initialization methods. Our implementation is based on Chen et al. (2021c).

Under the real world setting, we do not expect our theorem to hold *exactly*. Instead, our theorem implies that (1) there exists a threshold such that the testing performance is no much worse than (or sometimes may slightly better than) its dense counter part; and (2) the training error decreases later than the testing error decreases. Our experiments on MLP (Figure 3a) and VGG-16 (Figure 3b) show that this is the case: for MLP the test accuracy is steady competitive to its dense counterpart when the sparsity is less than $79\%$ and $36\%$ for VGG-16. We further provide experiments on ResNet in the appendix for validation of our theoretical results.

## 6 DISCUSSION AND FUTURE DIRECTION

In this work, we provide theory on the generalization performance of pruned neural networks trained by gradient descent under different pruning rates. Our results characterize the effect of pruning under different pruning rates: in the mild pruning case, the signal in the feature is well-preserved and the noise level is reduced which leads to improvement in the trained network's generalization; on the other hand, over pruning significantly destroys signal strength despite of reducing noise variance. One open problem on this topic still appears challenging. In this paper, we characterize two cases of pruning: in mild pruning the signal is preserved and in over pruning the signal is completely destroyed. However, the transition between these two cases is not well-understood. Further, it would be interesting to consider more general data distribution, and understand how pruning affects training multi-layer neural networks. We leave these interesting directions as future works.

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

## A    EXPERIMENT DETAILS

The experiments of MLP, VGG and ResNet-32 are run on NVIDIA A5000 and ResNet-50 and ResNet-20-128 is run on 4 NIVIDIA V100s. We list the hyperparameters we used in training. All of our models are trained with SGD and the detailed settings are summarized below.

Table 1: Summary of architectures, dataset and training hyperparameters

| MODEL | DATA | EPOCH | BATCH SIZE | LR | MOMENTUM | LR DECAY, EPOCH | WEIGHT DECAY |
|---|---|---|---|---|---|---|---|
| LENET | MNIST | 120 | 128 | 0.1 | 0 | 0 | 0 |
| VGG | CIFAR-10 | 160 | 128 | 0.1 | 0.9 | $0.1 \times [80, 120]$ | 0.0001 |
| RESNETS | CIFAR-10 | 160 | 128 | 0.1 | 0.9 | $0.1 \times [80, 120]$ | 0.0001 |

## B    FURTHER EXPERIMENT RESULTS

We plot the experiment result of ResNet-20-128 in Figure 4. This figure further verifies our results that there exists pruning rate threshold such that the testing performance of the pruned network is on par with the testing performance of the dense model while the training accuracy remains perfect.

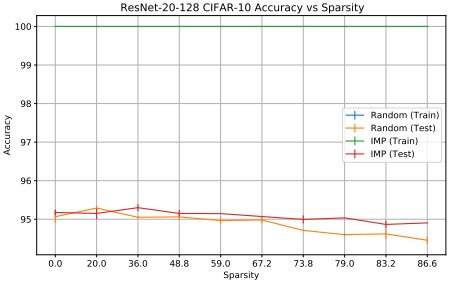

Figure 4: The figure shows the experiment results of ResNet-20-128 under various sparsity by random pruning and IMP. Each data point is averaged over 2 runs.

## C    PRELIMINARY FOR ANALYSIS

In this section, we introduce the following signal-noise decomposition of each neuron weight from Cao et al. (2022), and some useful properties for the terms in such a decomposition, which are useful in our analysis.

**Definition C.1** (signal-noise decomposition). *For each neuron weight $j \in [K]$, $r \in [m]$, there exist coefficients $\gamma_{j,r,k}^{(t)}, \zeta_{j,r,i}^{(t)}, \omega_{j,r,i}^{(t)}$ such that*

$$\widetilde{\mathbf{w}}_{j,r}^{(t)} = \widetilde{\mathbf{w}}_{j,r}^{(0)} + \sum_{k=1}^{K} \gamma_{j,r,k}^{(t)} \cdot \|\boldsymbol{\mu}_k\|_2^{-2} \cdot \boldsymbol{\mu}_k \odot \mathbf{m}_{j,r} + \sum_{i=1}^{n} \zeta_{j,r,i}^{(t)} \cdot \left\|\widetilde{\boldsymbol{\xi}}_{j,r,i}\right\|_2^{-2} \cdot \widetilde{\boldsymbol{\xi}}_{j,r,i} + \sum_{i=1}^{n} \omega_{j,r,i}^{(t)} \left\|\widetilde{\boldsymbol{\xi}}_{j,r,i}\right\|_2^{-2} \cdot \widetilde{\boldsymbol{\xi}}_{j,r,i},$$

*where $\gamma_{j,r,j}^{(t)} \geq 0$, $\gamma_{j,r,k}^{(t)} \leq 0$, $\zeta_{j,r,i}^{(t)} \geq 0, \omega_{j,r,i}^{(t)} \leq 0$.*

It is straightforward to see the following:

$$\gamma_{j,r,k}^{(0)}, \zeta_{j,r,i}^{(0)}, \omega_{j,r,i}^{(0)} = 0,$$

$$\gamma_{j,r,j}^{(t+1)} = \gamma_{j,r,j}^{(t)} - \mathbb{I}(r \in \mathcal{S}_{\text{signal}}^j) \frac{\eta}{n} \sum_{i=1}^{n} \ell_{j,i}'^{(t)} \cdot \sigma'\left(\left\langle \widetilde{\mathbf{w}}_{j,r}^{(t)}, \boldsymbol{\mu}_{y_i} \right\rangle\right) \|\boldsymbol{\mu}_{y_i}\|_2^2 \, \mathbb{I}(y_i = j),$$

$$\gamma_{j,r,k}^{(t+1)} = \gamma_{j,r,k}^{(t)} - \mathbb{I}((\mathbf{m}_{j,r})_k = 1) \frac{\eta}{n} \sum_{i=1}^{n} \ell_{j,i}'^{(t)} \cdot \sigma'\left(\left\langle \widetilde{\mathbf{w}}_{j,r}^{(t)}, \boldsymbol{\mu}_{y_i} \right\rangle\right) \|\boldsymbol{\mu}_{y_i}\|_2^2 \, \mathbb{I}(y_i = k), \; \forall j \neq k,$$

$$\zeta_{j,r,i}^{(t+1)} = \zeta_{j,r,i}^{(t)} - \frac{\eta}{n} \cdot \ell_{j,i}'^{(t)} \cdot \sigma'\left(\left\langle \widetilde{\mathbf{w}}_{j,r}^{(t)}, \boldsymbol{\xi}_i \right\rangle\right) \left\|\widetilde{\boldsymbol{\xi}}_{j,r,i}\right\|_2^2 \mathbb{I}(j = y_i),$$

$$\omega_{j,r,i}^{(t+1)} = \omega_{j,r,i}^{(t)} - \frac{\eta}{n} \cdot \ell_{j,i}'^{(t)} \cdot \sigma'\left(\left\langle \widetilde{\mathbf{w}}_{j,r}^{(t)}, \boldsymbol{\xi}_i \right\rangle\right) \left\|\widetilde{\boldsymbol{\xi}}_{j,r,i}\right\|_2^2 \mathbb{I}(j \neq y_i),$$

where $\{\gamma_{j,r,j}^{(t)}\}_{t=1}^{T}, \{\zeta_{j,r,i}^{(t)}\}_{t=1}^{T}$ are increasing sequences and $\{\gamma_{j,r,k}^{(t)}\}_{t=1}^{T}, \{\omega_{j,r,i}^{(t)}\}_{t=1}^{T}$ are decreasing sequences, because $-\ell_{j,i}'^{(t)} \geq 0$ when $j = y_i$, and $-\ell_{j,i}'^{(t)} \leq 0$ when $j \neq y_i$. By Lemma D.4, we have $pd > n + K$, and hence the set of vectors $\{\boldsymbol{\mu}_k\}_{k=1}^{K} \bigcup \{\widetilde{\boldsymbol{\xi}}_i\}_{i=1}^{n}$ is linearly independent with probability measure 1 over the Gaussian distribution for each $j \in [K], r \in [m]$. Therefore the decomposition is unique.

## D   PROOF OF THEOREM 3.1

We first formally restate Theorem 3.1.

**Theorem D.1** (Formal Restatement of Theorem 3.1). *Under Condition 2.2, choose initialization variance $\sigma_0 = \widetilde{\Theta}(m^{-4} n^{-1} \mu^{-1})$ and learning rate $\eta \leq \widetilde{O}(1/\mu^2)$. For $\epsilon > 0$, if $p \geq C_1 \frac{\log d}{m}$ for some sufficiently large constant $C_1$, then with probability at least $1 - O(d^{-1})$ over the randomness in the data, network initialization and pruning, there exists $T = \widetilde{O}(K\eta^{-1} \sigma_0^{2-q} \mu^{-q} + K^2 m^4 \mu^{-2} \eta^{-1} \epsilon^{-1})$ such that the following holds:*

1. *The training loss is below $\epsilon$: $L_S(\widetilde{\mathbf{W}}^{(T)}) \leq \epsilon$.*

2. *The weights of the CNN highly correlate with its corresponding class signal: $\max_r \gamma_{j,r,j}^{(T)} \geq \Omega(m^{-1/q})$ for all $j \in [K]$.*

3. *The weights of the CNN doesn't have high correlation with the signal from different classes: $\max_{j \neq k, r \in [m]} |\gamma_{j,r,k}^{(T)}| \leq \widetilde{O}(\sigma_0 \mu)$.*

4. *None of the weights is highly correlated with the noise: $\max_{j,r,i} \zeta_{j,r,i}^{(T)} = \widetilde{O}(\sigma_0 \sigma_n \sqrt{pd})$, $\max_{j,r,i} |\omega_{j,r,i}^{(T)}| = \widetilde{O}(\sigma_0 \sigma_n \sqrt{pd})$.*

*Moreover, the testing loss is upper-bounded by*

$$L_{\mathcal{D}}(\widetilde{\mathbf{W}}^{(T)}) \leq O(K\epsilon) + \exp(-n^2/p).$$

The proof of Theorem 3.1 consists of the analysis of the pruning on the signal and noise for three stages of gradient descent: initialization, feature growing phase, and converging phase, and the establishment of the generalization property. We present these analysis in detail in the following subsections. A special note is that the constant $C$ showing up in the following proof of each subsequent Lemmas is defined locally instead of globally, which means the constant $C$ within each Lemma is the same but may be different across different Lemma.

### D.1   INITIALIZATION

We analyze the effect of pruning on weight-signal correlation and weight-noise correlation at the initialization. We first present a few supporting lemmas, and finally provide our main result of

Lemma D.7, which shows that if the pruning is mild, then it will not hurt the max weight-signal correlation much at the initialization. On the other hand, the max weight-noise correlation is reduced by a factor of $\sqrt{p}$.

**Lemma D.2.** *Assume $n = \Omega(K^2 \log Kd)$. Then, with probability at least $1 - 1/d$,*

$$|\{i \in [n] : y_i = j\}| = \Theta(n/K) \quad \forall j \in [K].$$

*Proof.* By Hoeffding's inequality, with probability at least $1 - \delta/2K$, for a fixed $j \in [K]$, we have

$$\left| \frac{1}{n} \sum_{i=1}^{n} \mathbb{I}(y_i = j) - \frac{1}{K} \right| \leq \sqrt{\frac{\log(4K/\delta)}{2n}}.$$

Therefore, as long as $n \geq 2K^2 \log(4K/\delta)$, we have

$$\left| \frac{1}{n} \sum_{i=1}^{n} \mathbb{I}(y_i = j) - \frac{1}{K} \right| \leq \frac{1}{2K}.$$

Taking a union bound over $j \in [K]$ and making $\delta = 1/d$ yield the result. $\qquad\square$

**Lemma D.3.** *Assume $pm = \Omega(\log d)$ and $m = \operatorname{poly} \log d$. Then, with probability $1 - 1/d$, for all $j \in [K]$, $k \in [K]$, we have $\sum_{r=1}^{m} (\mathbf{m}_{j,r})_k = \Theta(pm)$, which implies that $|\mathcal{S}_{\text{signal}}^{j}| = \Theta(pm)$ for all $j \in [K]$.*

*Proof.* When $pm = \Omega(\log d)$, by multiplicative Chernoff's bound, for a given $k \in [K]$, we have

$$\mathbb{P}\left[ \left| \sum_{r=1}^{m} (\mathbf{m}_{j,r})_k - pm \right| \geq 0.5pm \right] \leq 2 \exp\{-\Omega(pm)\}.$$

Take a union bound over $j \in [K]$, $k \in [K]$, we have

$$\mathbb{P}\left[ \left| \sum_{r=1}^{m} (\mathbf{m}_{j,r})_k - pm \right| \geq 0.5pm, \ \forall j \in [K], \ k \in [K] \right] \leq 2K^2 \exp\{-\Omega(pm)\} \leq 1/d.$$

$\qquad\square$

**Lemma D.4.** *Assume $p = 1/\operatorname{poly} \log d$. Then with probability at least $1 - 1/d$, for all $j \in [K]$, $r \in [m]$, $\sum_{i=1}^{d} (\mathbf{m}_{j,r})_i = \Theta(pd)$.*

*Proof.* By multiplicative Chernoff's bound, we have for a given $j, r$

$$\mathbb{P}\left[ \left| \sum_{i=1}^{d} (\mathbf{m}_{j,r})_i - pd \right| \geq 0.5pd \right] \leq 2 \exp\{-\Omega(pd)\}.$$

Take a union bound over $j, r$, we have

$$\mathbb{P}\left[ \left| \sum_{i=1}^{d} (\mathbf{m}_{j,r})_i - pd \right| \geq 0.5pd, \ \forall j \in [K], \ r \in [m] \right] \leq 2Km \exp\{-\Omega(pd)\} \leq 1/d,$$

where the last inequality follows from our choices of $p, K, m, d$. $\qquad\square$

**Lemma D.5.** *Suppose $p = \Omega(1/\operatorname{poly} \log d)$, and $m, n = \operatorname{poly} \log d$. With probability at least $1 - 1/d$, we have*

$$\left\| \widetilde{\boldsymbol{\xi}}_{j,r,i} \right\|_2^2 = \Theta(\sigma_n^2 pd),$$

$$\left| \left\langle \widetilde{\boldsymbol{\xi}}_{j,r,i}, \boldsymbol{\xi}_{i'} \right\rangle \right| \leq O(\sigma_n^2 \sqrt{pd \log d}),$$

$$\left| \left\langle \boldsymbol{\mu}_k, \widetilde{\boldsymbol{\xi}}_{j,r,i} \right\rangle \right| \leq |\langle \boldsymbol{\mu}, \boldsymbol{\xi}_i \rangle| \leq O(\sigma_n \mu \sqrt{\log d}),$$

*for all $j \in \{-1, 1\}$, $r \in [m]$, $i, i' \in [n]$ and $i \neq i'$.*

*Proof.* From Lemma D.4, we have with probability at least $1 - 1/d$,

$$\sum_{k=1}^{d} (\mathbf{m}_{j,r})_k = \Theta(pd), \quad \forall j \in [K], \ r \in [m].$$

For a set of Gaussian random variable $g_1, \ldots, g_N \sim \mathcal{N}(0, \sigma^2)$, by Bernstein's inequality, with probability at least $1 - \delta$, we have

$$\left| \sum_{i=1}^{N} g_i^2 - \sigma^2 N \right| \lesssim \sigma^2 \sqrt{N \log \frac{1}{\delta}}.$$

Thus, by a union bound over $j, r, i$, with probability at least $1 - 1/d$, we have

$$\left\| \widetilde{\boldsymbol{\xi}}_{j,r,i} \right\|_2^2 = \Theta(\sigma_n^2 pd).$$

For $i \neq i'$, again by Bernstein's bound, we have with probability at least $1 - \delta$,

$$\left| \left\langle \widetilde{\boldsymbol{\xi}}_{j,r,i}, \boldsymbol{\xi}_{i'} \right\rangle \right| \leq O\left( \sigma_n^2 \sqrt{pd \log \frac{Kmn}{\delta}} \right),$$

for all $j, r, i$. Plugging in $\delta = 1/d$ gives the result. The proof for $|\langle \boldsymbol{\mu}, \boldsymbol{\xi}_i \rangle|$ is similar. $\qquad \square$

**Lemma D.6.** *Suppose we have $m$ independent Gaussian random variables $g_1, g_2, \ldots, g_m \sim \mathcal{N}(0, \sigma^2)$. Then with probability $1 - \delta$,*

$$\max_i g_i \geq \sigma \sqrt{\log \frac{m}{\log 1/\delta}}.$$

*Proof.* By the standard tail bound of Gaussian random variable, we have for every $x > 0$,

$$\left( \frac{\sigma}{x} - \frac{\sigma^3}{x^3} \right) \frac{e^{-x^2/2\sigma^2}}{\sqrt{2\pi}} \leq \mathbb{P}\left[ g > x \right] \leq \frac{\sigma}{x} \frac{e^{-x^2/2\sigma^2}}{\sqrt{2\pi}}.$$

We want to pick a $x^\star$ such that

$$\mathbb{P}\left[ \max_i g_i \leq x^\star \right] = (\mathbb{P}\left[ g_i \leq x^\star \right])^m = (1 - \mathbb{P}\left[ g_i \geq x^\star \right])^m \leq e^{-m \, \mathbb{P}[g_i \geq x^\star]} \leq \delta$$

$$\Rightarrow \mathbb{P}[g_i \geq x^\star] = \Theta\left( \frac{\log(1/\delta)}{m} \right)$$

$$\Rightarrow x^\star = \Theta(\sigma \sqrt{\log(m/(\log(1/\delta) \log m))}).$$

$\qquad \square$

**Lemma D.7** (Formal Restatement of Lemma 3.2). *With probability at least $1 - 2/d$, for all $i \in [n]$,*

$$\sigma_0 \sigma_n \sqrt{pd} \leq \max_r \left\langle \widetilde{\mathbf{w}}_{j,r}^{(0)}, \boldsymbol{\xi}_i \right\rangle \leq \sqrt{2 \log(Kmd)} \sigma_0 \sigma_n \sqrt{pd}.$$

*Further, suppose $pm \geq \Omega(\log(Kd))$. Then with probability $1 - 2/d$, for all $j \in [K]$,*

$$\sigma_0 \left\| \boldsymbol{\mu}_j \right\|_2 \leq \max_{r \in \mathcal{S}_{\text{signal}}^j} \left\langle \widetilde{\mathbf{w}}_{j,r}^{(0)}, \boldsymbol{\mu}_j \right\rangle \leq \sqrt{2 \log(8pmKd)} \sigma_0 \left\| \boldsymbol{\mu}_j \right\|_2.$$

*Proof.* We first give a proof for the second inequality. From Lemma D.3, we know that $|\mathcal{S}_{\text{signal}}^j| = \Theta(pm)$. The upper bound can be obtained by taking a union bound over $r \in \mathcal{S}_{\text{signal}}^j$, $j \in [K]$. To prove the lower bound, applying Lemma D.6, with probability at least $1 - \delta/K$, we have for a given $j \in [K]$

$$\max_{r \in \mathcal{S}_{\text{signal}}^j} \left\langle \widetilde{\mathbf{w}}_{j,r}^{(0)}, \boldsymbol{\mu}_j \right\rangle \geq \sigma_0 \left\| \boldsymbol{\mu}_j \right\|_2 \sqrt{\log \frac{pm}{\log K/\delta}}.$$

Now, notice that we can control the constant in $pm$ (by controlling the constant in the lower bound of $p$) such that $pm/\log(Kd) \geq e$. Thus, taking a union bound over $j \in [K]$ and setting $\delta = 1/d$ yield the result.

The proof of the first inequality is similar. $\qquad \square$

### D.2    SUPPORTING PROPERTIES FOR ENTIRE TRAINING PROCESS

This subsection establishes a few properties (summarized in Proposition D.10) that will be used in the analysis of feature growing phase and converging phase of gradient descent presented in the next two subsections. Define $T^\star = \eta^{-1} \text{poly}(1/\epsilon, \mu, d^{-1}, \sigma_n^{-2}, \sigma_0^{-1} n, m, d)$. Denote $\alpha = \Theta(\log^{1/q}(T^\star))$, $\beta = 2\max_{i,j,r,k}\left\{\left|\left\langle \widetilde{\mathbf{w}}_{j,r}^{(0)}, \boldsymbol{\mu}_k \right\rangle\right|, \left|\left\langle \widetilde{\mathbf{w}}_{j,r}^{(0)}, \boldsymbol{\xi}_i \right\rangle\right|\right\}$. We need the following bound holds for our subsequent analysis.

$$4m^{1/q}\max_{j,r,i}\left\{\left\langle \widetilde{\mathbf{w}}_{j,r}^{(0)}, \boldsymbol{\mu}_{y_i} \right\rangle, Cn\alpha\frac{\mu\sqrt{\log d}}{\sigma_n pd}, \left\langle \widetilde{\mathbf{w}}_{j,r}^{(0)}, \boldsymbol{\xi}_i \right\rangle, 3Cn\alpha\sqrt{\frac{\log d}{pd}}\right\} \leq 1 \qquad (1)$$

**Remark D.8.** To see why Equation (1) can hold under Condition 2.2, we convert everything in terms of $d$. First recall from Condition 2.2 that $m, n = \text{poly}(\log d)$ and $\mu = \Theta(\sigma_n\sqrt{d}\log d) = \Theta(1)$. In both mild pruning and over pruning we require $p \geq \Omega(1/\text{poly}\log d)$. Since $\alpha = \Theta(\log^{1/q}(T^\star))$, if we assume $T^\star \leq O(\text{poly}(d))$ for a moment (which we are going to justify in the next paragraph), then $\alpha = O(\log^{1/q}(d))$. Then if we set $d$ to be large enough, we have $4m^{1/q}Cn\alpha\frac{\mu\sqrt{\log d}}{\sigma_n pd} \leq \frac{\text{poly}\log d}{\sqrt{d}} \leq 1$. Finally for the quantity $4m^{1/q}\max_{j,r,i}\{\langle\widetilde{\mathbf{w}}_{j,r}^{(0)}, \boldsymbol{\mu}_{y_i}\rangle, \langle\widetilde{\mathbf{w}}_{j,r}^{(0)}, \boldsymbol{\xi}_i\rangle\}$, by Lemma 3.2, our assumption of $K = O(\log d)$ in Condition 2.2 and our choice of $\sigma_0 = \widetilde{\Theta}(m^{-4}n^{-1}\mu^{-1})$ in Theorem 3.1 (or Theorem D.1), we can easily see that this quantity can also be made smaller than 1.

Now, to justify that $T^\star \leq O(\text{poly}(d))$, we only need to justify that all the quantities $T^\star$ depend on is polynomial in $d$. First of all, based on Condition 2.2, $n, m = \text{poly}\log(d)$ and $\mu = \Theta(\sigma_n\sqrt{d}\log d) = \Theta(1)$ further implies $\sigma_n^{-2} = \Theta(d\log^2 d)$. Since Theorem 3.1 only requires $\sigma_0 = \widetilde{\Theta}(m^{-4}n^{-1}\mu^{-1})$, this implies $\sigma_0^{-1} \leq O(\text{poly}\log d)$. Hence $\sigma_0^{-1}n = O(\text{poly}\log d)$. Together with our assumption that $\epsilon, \eta \geq \Omega(1/\text{poly}(d))$ (which implies $1/\epsilon, 1/\eta \leq O(\text{poly}(d))$), we have justified that all terms involved in $T^\star$ are at most of order $\text{poly}(d)$. Hence $T^\star = \text{poly}(d)$.

**Remark D.9.** Here we make remark on our assumption on $\epsilon$ and $\eta$ in Condition 2.2.

For our assumption on $\epsilon$, since the cross-entropy loss is (1) not strongly-convex and (2) achieves its infimum at infinity. In practice, the cross-entropy loss is minimized to a constant level, say 0.001. We make this assumption to avoid the pathological case where $\epsilon$ is exponentially small in $d$ (say $\epsilon = 2^{-d}$) which is unrealistic. Thus, for realistic setting, we assume $\epsilon \geq \Omega(1/\text{poly}(d))$ or $1/\epsilon \leq O(\text{poly}(d))$.

To deal with $\eta$, the only restriction we have is $\eta = O(1/\mu^2)$ in Theorem 3.1 and Theorem 4.1. However, in practice, we don't use a learning rate that is exponentially small, say $\eta = 2^{-d}$. Thus, like dealing with $\epsilon$, we assume $\eta \geq \Omega(1/\text{poly}(d))$ or $1/\eta \leq O(\text{poly}\, d)$.

We make the above assumption to simplify analysis when analyzing the magnitude of $F_j(X)$ for $j \neq y$ given sample $(X, y)$.

**Proposition D.10.** *Under Condition 2.2, during the training time $t < T^\star$, we have*

1. $\gamma_{j,r,j}^{(t)}, \zeta_{j,r,i}^{(t)} \leq \alpha,$

2. $\omega_{j,r,i}^{(t)} \geq -\beta - 6Cn\alpha\sqrt{\frac{\log d}{pd}}.$

3. $\gamma_{j,r,k}^{(t)} \geq -\beta - 2Cn\alpha\frac{\mu\sqrt{\log d}}{\sigma_n pd}.$

*Notice that the lower bound has absolute value smaller than the upper bound.*

*Proof of Proposition D.10.* We use induction to prove Proposition D.10.

**Induction Hypothesis:**    Suppose Proposition D.10 holds for all $t < T \leq T^\star$.

We next show that this also holds for $t = T$ via the following a few lemmas.

**Lemma D.11.** *Under Condition 2.2, for $t < T$, there exists a constant $C$ such that*

$$\left\langle \widetilde{\mathbf{w}}_{j,r}^{(t)} - \widetilde{\mathbf{w}}_{j,r}^{(0)}, \boldsymbol{\mu}_k \right\rangle = \left( \gamma_{j,r,k}^{(t)} \pm Cn\alpha \frac{\mu\sqrt{\log d}}{\sigma_n pd} \right) \mathbb{I}((\mathbf{m}_{j,r})_k = 1),$$

$$\left\langle \widetilde{\mathbf{w}}_{j,r}^{(t)} - \widetilde{\mathbf{w}}_{j,r}^{(0)}, \boldsymbol{\xi}_i \right\rangle = \zeta_{j,r,i}^{(t)} \pm 3Cn\alpha\sqrt{\frac{\log d}{pd}},$$

$$\left\langle \widetilde{\mathbf{w}}_{j,r}^{(t)} - \widetilde{\mathbf{w}}_{j,r}^{(0)}, \boldsymbol{\xi}_i \right\rangle = \omega_{j,r,i}^{(t)} \pm 3Cn\alpha\sqrt{\frac{\log d}{pd}}.$$

*Proof.* From Lemma D.5, there exists a constant $C$ such that with probability at least $1 - 1/d$,

$$\frac{\left| \left\langle \widetilde{\boldsymbol{\xi}}_{j,r,i}, \boldsymbol{\xi}_{i'} \right\rangle \right|}{\left\| \widetilde{\boldsymbol{\xi}}_{j,r,i} \right\|_2^2} \leq C\sqrt{\frac{\log d}{pd}},$$

$$\frac{\left| \left\langle \widetilde{\boldsymbol{\xi}}_{j,r,i}, \boldsymbol{\mu}_k \right\rangle \right|}{\left\| \widetilde{\boldsymbol{\xi}}_{j,r,i} \right\|_2^2} \leq C\frac{\mu\sqrt{\log d}}{\sigma_n pd},$$

$$\frac{\left| \left\langle \boldsymbol{\mu}_k, \boldsymbol{\xi}_i \right\rangle \right|}{\left\| \boldsymbol{\mu}_k \right\|_2^2} \leq C\frac{\sigma_n\sqrt{\log d}}{\mu}.$$

Using the signal-noise decomposition and assuming $(\mathbf{m}_{j,r})_k = 1$, we have

$$\left| \left\langle \widetilde{\mathbf{w}}_{j,r}^{(t)} - \widetilde{\mathbf{w}}_{j,r}^{(0)}, \boldsymbol{\mu}_k \right\rangle - \gamma_{j,r,k}^{(t)} \right| = \left| \sum_{i=1}^n \zeta_{j,r,i}^{(t)} \cdot \left\| \widetilde{\boldsymbol{\xi}}_{j,r,i} \right\|_2^{-2} \cdot \left\langle \widetilde{\boldsymbol{\xi}}_{j,r,i}, \boldsymbol{\mu}_k \right\rangle + \sum_{i=1}^n \omega_{j,r,i}^{(t)} \left\| \widetilde{\boldsymbol{\xi}}_{j,r,i} \right\|_2^{-2} \cdot \left\langle \widetilde{\boldsymbol{\xi}}_{j,r,i}, \boldsymbol{\mu}_k \right\rangle \right|$$

$$\leq C\frac{\mu\sqrt{\log d}}{\sigma_n pd} \sum_{i=1}^n \left| \zeta_{j,r,i}^{(t)} \right| + C\frac{\mu\sqrt{\log d}}{\sigma_n pd} \sum_{i=1}^n \left| \omega_{j,r,i}^{(t)} \right|$$

$$\leq 2C\frac{\mu\sqrt{\log d}}{\sigma_n pd} n\alpha.$$

where the second last inequality is by Lemma D.5 and the last inequality is by induction hypothesis.

To prove the second equality, for $j = y_i$,

$$\left| \left\langle \widetilde{\mathbf{w}}_{j,r}^{(t)} - \widetilde{\mathbf{w}}_{j,r}^{(0)}, \boldsymbol{\xi}_i \right\rangle - \zeta_{j,r,i}^{(t)} \right| = \left| \sum_{k=1}^K \gamma_{j,r,k}^{(t)} \cdot \frac{\left\langle \boldsymbol{\mu}_k, \boldsymbol{\xi}_i \right\rangle}{\left\| \boldsymbol{\mu}_k \right\|_2^2} + \sum_{i' \neq i} \zeta_{j,r,i'}^{(t)} \cdot \frac{\left\langle \widetilde{\boldsymbol{\xi}}_{j,r,i'}, \boldsymbol{\xi}_i \right\rangle}{\left\| \widetilde{\boldsymbol{\xi}}_{j,r,i'} \right\|_2^2} + \sum_{i'=1}^n \omega_{j,r,i'}^{(t)} \frac{\left\langle \widetilde{\boldsymbol{\xi}}_{j,r,i'}, \boldsymbol{\xi}_i \right\rangle}{\left\| \widetilde{\boldsymbol{\xi}}_{j,r,i'} \right\|_2^2} \right|$$

$$\leq C\frac{\sigma_n\sqrt{\log d}}{\mu} \sum_{k=1}^K |\gamma_{j,r,k}^{(t)}| + C\sqrt{\frac{\log d}{pd}} \sum_{i' \neq i} |\zeta_{j,r,i'}^{(t)}| + C\sqrt{\frac{\log d}{pd}} \sum_{i'=1}^n |\omega_{j,r,i'}^{(t)}|$$

$$= C\frac{\sigma_n\sqrt{\log d}}{\mu} K\alpha + 2Cn\alpha\sqrt{\frac{\log d}{pd}}$$

$$\leq 3Cn\alpha\sqrt{\frac{\log d}{pd}}.$$

where the last inequality is by $n \gg K$ and $\mu = \Theta(\sigma_n\sqrt{d}\log d)$. The proof for the case of $j \neq y_i$ is similar. $\qquad\square$

**Lemma D.12** (Off-diagonal Correlation Upper Bound). *Under Condition 2.2, for $t < T$, $j \neq y_i$, we have that*

$$\left\langle \widetilde{\mathbf{w}}_{j,r}^{(t)}, \boldsymbol{\mu}_{y_i} \right\rangle \leq \left\langle \widetilde{\mathbf{w}}_{j,r}^{(0)}, \boldsymbol{\mu}_{y_i} \right\rangle + Cn\alpha \frac{\mu\sqrt{\log d}}{\sigma_n pd},$$

$$\left\langle \widetilde{\mathbf{w}}_{j,r}^{(t)}, \boldsymbol{\xi}_i \right\rangle \leq \left\langle \widetilde{\mathbf{w}}_{j,r}^{(0)}, \boldsymbol{\xi}_i \right\rangle + 3Cn\alpha\sqrt{\frac{\log d}{pd}},$$

$$F_j(\widetilde{\mathbf{W}}_j^{(t)}, \mathbf{x}_i) \leq 1.$$

*Proof.* If $j \neq y_i$, then $\gamma_{j,r,k}^{(t)} \leq 0$ and we have that

$$\left\langle \widetilde{\mathbf{w}}_{j,r}^{(t)}, \boldsymbol{\mu}_{y_i} \right\rangle \leq \left\langle \widetilde{\mathbf{w}}_{j,r}^{(0)}, \boldsymbol{\mu}_{y_i} \right\rangle + \left( \gamma_{j,r,y_i}^{(t)} + Cn\alpha \frac{\mu\sqrt{\log d}}{\sigma_n pd} \right) \mathbb{I}((\mathbf{m}_{j,r})_{y_i} = 1)$$

$$\leq \left\langle \widetilde{\mathbf{w}}_{j,r}^{(0)}, \boldsymbol{\mu}_{y_i} \right\rangle + Cn\alpha \frac{\mu\sqrt{\log d}}{\sigma_n pd}.$$

Further, we can obtain

$$\left\langle \widetilde{\mathbf{w}}_{j,r}^{(t)}, \boldsymbol{\xi}_i \right\rangle \leq \left\langle \widetilde{\mathbf{w}}_{j,r}^{(0)}, \boldsymbol{\xi}_i \right\rangle + \omega_{j,r,i}^{(t)} + 3Cn\alpha\sqrt{\frac{\log d}{pd}}$$

$$\leq \left\langle \widetilde{\mathbf{w}}_{j,r}^{(0)}, \boldsymbol{\xi}_i \right\rangle + 3Cn\alpha\sqrt{\frac{\log d}{pd}}.$$

Then, we have the following bound:

$$F_j(\widetilde{\mathbf{W}}_j^{(t)}, \mathbf{x}_i) = \sum_{r=1}^{m} [\sigma(\langle \widetilde{\mathbf{w}}_{j,r}, \boldsymbol{\mu}_{y_i} \rangle) + \sigma(\langle \widetilde{\mathbf{w}}_{j,r}, \boldsymbol{\xi}_i \rangle)]$$

$$\leq m2^{q+1} \max_{j,r,i} \left\{ \left\langle \widetilde{\mathbf{w}}_{j,r}^{(0)}, \boldsymbol{\mu}_{y_i} \right\rangle, Cn\alpha \frac{\mu\sqrt{\log d}}{\sigma_n pd}, \left\langle \widetilde{\mathbf{w}}_{j,r}^{(0)}, \boldsymbol{\xi}_i \right\rangle, 3Cn\alpha\sqrt{\frac{\log d}{pd}} \right\}^q$$

$$\leq 1.$$

where the first inequality is by Equation (1). $\qquad\qquad\square$

**Lemma D.13** (Diagonal Correlation Upper Bound). *Under Condition 2.2, for $t < T$, $j = y_i$, we have*

$$\left\langle \widetilde{\mathbf{w}}_{j,r}^{(t)}, \boldsymbol{\mu}_j \right\rangle \leq \left\langle \widetilde{\mathbf{w}}_{j,r}^{(0)}, \boldsymbol{\mu}_j \right\rangle + \gamma_{j,r,j}^{(t)} + Cn\alpha \frac{\mu\sqrt{\log d}}{\sigma_n pd},$$

$$\left\langle \widetilde{\mathbf{w}}_{j,r}^{(t)}, \boldsymbol{\xi}_i \right\rangle \leq \left\langle \widetilde{\mathbf{w}}_{j,r}^{(0)}, \boldsymbol{\xi}_i \right\rangle + \zeta_{j,r,i}^{(t)} + 3Cn\alpha\sqrt{\frac{\log d}{pd}}.$$

*If $\max\{\gamma_{j,r,j}^{(t)}, \zeta_{j,r,i}^{(t)}\} \leq m^{-1/q}$, we further have that $F_j(\widetilde{\mathbf{W}}_j^{(t)}, \mathbf{x}_i) \leq O(1)$.*

*Proof.* The two inequalities are immediate consequences of Lemma D.11. If $\max\{\gamma_{j,r,j}^{(t)}, \zeta_{j,r,i}^{(t)}\} \leq m^{-1/q}$, we have

$$F_j(\widetilde{\mathbf{W}}_j^{(t)}, \mathbf{x}_i) = \sum_{r=1}^{m} [\sigma(\langle \widetilde{\mathbf{w}}_{j,r}, \boldsymbol{\mu}_j \rangle) + \sigma(\langle \widetilde{\mathbf{w}}_{j,r}, \boldsymbol{\xi}_i \rangle)]$$

$$\leq 2 \cdot 3^q m \max_{j,r,i} \left\{ \gamma_{j,r}^{(t)}, \zeta_{j,r,i}^{(t)}, \left| \left\langle \widetilde{\mathbf{w}}_{j,r}^{(0)}, \boldsymbol{\mu}_j \right\rangle \right|, \left| \left\langle \widetilde{\mathbf{w}}_{j,r}^{(0)}, \boldsymbol{\xi}_i \right\rangle \right|, Cn\alpha \frac{\mu\sqrt{\log d}}{\sigma_n pd}, 3Cn\alpha\sqrt{\frac{\log d}{pd}} \right\}^q$$

$$\leq O(1).$$

$\qquad\qquad\square$

**Lemma D.14.** *Under Condition 2.2, for $t \leq T$, we have that*

1. $\omega_{j,r,i}^{(t)} \geq -\beta - 6Cn\alpha\sqrt{\frac{\log d}{pd}}$;

2. $\gamma_{j,r,k}^{(t)} \geq -\beta - 2Cn\alpha\frac{\mu\sqrt{\log d}}{\sigma_n pd}$.

*Proof.* When $j = y_i$, we have $\omega_{j,r,i}^{(t)} = 0$. We only need to consider the case of $j \neq y_i$. When $\omega_{j,r,i}^{(T-1)} \leq -0.5\beta - 3Cn\alpha\sqrt{\frac{\log d}{pd}}$, by Lemma D.11 we have

$$\left\langle \widetilde{\mathbf{w}}_{j,r}^{(T-1)}, \boldsymbol{\xi}_i \right\rangle \leq \left\langle \widetilde{\mathbf{w}}_{j,r}^{(0)}, \boldsymbol{\xi}_i \right\rangle + \omega_{j,r,i}^{(T-1)} + 3Cn\alpha\sqrt{\frac{\log d}{pd}} \leq 0.$$

Thus,

$$\begin{aligned}
\omega_{j,r,i}^{(T)} &= \omega_{j,r,i}^{(T-1)} - \frac{\eta}{n} \cdot \ell_{j,i}'^{(T-1)} \cdot \sigma' \left( \left\langle \widetilde{\mathbf{w}}_{j,r}^{(T-1)}, \boldsymbol{\xi}_i \right\rangle \right) \left\| \widetilde{\boldsymbol{\xi}}_{j,r,i} \right\|_2^2 \mathbb{I}(j \neq y_i) \\
&= \omega_{j,r,i}^{(T-1)} \\
&\geq -\beta - 6Cn\alpha\sqrt{\frac{\log d}{pd}}.
\end{aligned}$$

When $\omega_{j,r,i}^{(T-1)} \geq -0.5\beta - 3Cn\alpha\sqrt{\frac{\log d}{pd}}$, we have

$$\begin{aligned}
\omega_{j,r,i}^{(T)} &= \omega_{j,r,i}^{(T-1)} - \frac{\eta}{n} \cdot \ell_{j,i}'^{(T-1)} \cdot \sigma' \left( \left\langle \widetilde{\mathbf{w}}_{j,r}^{(T-1)}, \boldsymbol{\xi}_i \right\rangle \right) \left\| \widetilde{\boldsymbol{\xi}}_{j,r,i} \right\|_2^2 \mathbb{I}(j \neq y_i) \\
&\geq -0.5\beta - 3Cn\alpha\sqrt{\frac{\log d}{pd}} - \frac{\eta}{n}\sigma' \left( 0.5\beta + 3Cn\alpha\sqrt{\frac{\log d}{pd}} \right) \left\| \widetilde{\boldsymbol{\xi}}_{j,r,i} \right\|_2^2 \\
&\geq -\beta - 6Cn\alpha\sqrt{\frac{\log d}{pd}},
\end{aligned}$$

where the last inequality is by setting $\eta \leq nq^{-1}\left(0.5\beta + 3Cn\alpha\sqrt{\frac{\log d}{pd}}\right)^{2-q}(C_2\sigma_n^2 d)^{-1}$ and $C_2$ is the constant such that $\left\| \widetilde{\boldsymbol{\xi}}_{j,r,i} \right\|_2^2 \leq C_2\sigma_n^2 pd$ for all $j, r, i$ in Lemma D.5.

For $\gamma_{j,r,k}^{(t)}$, the proof is similar. Consider $\mathbb{I}((\mathbf{m}_{j,r})_k) = 1$. When $\gamma_{j,r,k}^{(t)} \leq -0.5\beta - Cn\alpha\frac{\mu\sqrt{\log d}}{\sigma_n pd}$, by Lemma D.11, we have

$$\left\langle \widetilde{\mathbf{w}}_{j,r}^{(t)}, \boldsymbol{\mu}_k \right\rangle \leq \left\langle \widetilde{\mathbf{w}}_{j,r}^{(0)}, \boldsymbol{\mu}_k \right\rangle + \gamma_{j,r,k}^{(t)} + Cn\alpha\frac{\mu\sqrt{\log d}}{\sigma_n pd} \leq 0.$$

Hence,

$$\begin{aligned}
\gamma_{j,r,k}^{(T)} &= \gamma_{j,r,k}^{(T-1)} - \frac{\eta}{n}\sum_{i=1}^{n} \ell_{j,i}'^{(T-1)}\sigma' \left( \left\langle \widetilde{\mathbf{w}}_{j,r}^{(T-1)}, \boldsymbol{\mu}_k \right\rangle \right)\mu^2\mathbb{I}(y_i = k) \\
&= \gamma_{j,r,k}^{(T-1)} \\
&\geq -\beta - 2Cn\alpha\frac{\mu\sqrt{\log d}}{\sigma_n pd}.
\end{aligned}$$

When $\gamma_{j,r,k}^{(t)} \geq -0.5\beta - Cn\alpha \frac{\mu\sqrt{\log d}}{\sigma_n pd}$, we have

$$\begin{aligned}
\gamma_{j,r,k}^{(T)} &= \gamma_{j,r,k}^{(T-1)} - \frac{\eta}{n} \sum_{i=1}^{n} \ell_{j,i}'^{(T-1)} \sigma' \left( \left\langle \widetilde{\mathbf{w}}_{j,r}^{(T-1)}, \boldsymbol{\mu}_k \right\rangle \right) \mu^2 \mathbb{I}(y_i = k) \\
&\geq -0.5\beta - Cn\alpha \frac{\mu\sqrt{\log d}}{\sigma_n pd} - C_2 \frac{\eta}{K} \sigma' \left( 0.5\beta + Cn\alpha \frac{\mu\sqrt{\log d}}{\sigma_n pd} \right) \mu^2 \\
&\geq -\beta - 2Cn\alpha \frac{\mu\sqrt{\log d}}{\sigma_n pd},
\end{aligned}$$

where the first inequality follows from the fact that there are $\Theta(\frac{n}{K})$ samples such that $\mathbb{I}(y_i = k)$, and the last inequality follows from picking $\eta \leq K(0.5\beta + Cn\alpha \frac{\mu\sqrt{\log d}}{\sigma_n pd})^{2-q} \mu^{-2} q^{-1} C_2^{-1}$. $\qquad \square$

**Lemma D.15.** *Under Condition 2.2, for $t \leq T$, we have $\gamma_{j,r,j}^{(t)}, \zeta_{j,r,i}^{(t)} \leq \alpha$.*

*Proof.* For $y_i \neq j$ or $r \notin \mathcal{S}_{\text{signal}}^j$, $\gamma_{j,r,j}^{(t)}, \zeta_{j,r,i}^{(t)} = 0 \leq \alpha$.

If $y_i = j$, then by Lemma D.12 we have

$$\left| \ell_{j,i}'^{(t)} \right| = 1 - \text{logit}_j(F; X) = \frac{\sum_{i \neq j} e^{F_i(X)}}{\sum_{i=1}^{K} e^{F_i(X)}} \leq \frac{Ke}{e^{F_j(X)}}. \tag{2}$$

Recall that

$$\gamma_{j,r,j}^{(t+1)} = \gamma_{j,r,j}^{(t)} - \mathbb{I}(r \in \mathcal{S}_{\text{signal}}^j) \frac{\eta}{n} \sum_{i=1}^{n} \ell_{j,i}'^{(t)} \cdot \sigma' \left( \left\langle \widetilde{\mathbf{w}}_{j,r}^{(t)}, \boldsymbol{\mu}_{y_i} \right\rangle \right) \|\boldsymbol{\mu}_{y_i}\|_2^2 \mathbb{I}(y_i = j),$$

$$\zeta_{j,r,i}^{(t+1)} = \zeta_{j,r,i}^{(t)} - \frac{\eta}{n} \cdot \ell_{j,i}'^{(t)} \cdot \sigma' \left( \left\langle \widetilde{\mathbf{w}}_{j,r}^{(t)}, \boldsymbol{\xi}_i \right\rangle \right) \left\| \widetilde{\boldsymbol{\xi}}_{j,r,i} \right\|_2^2 \mathbb{I}(j = y_i).$$

We first bound $\zeta_{j,r,i}^{(T)}$. Let $T_{j,r,i}$ be the last time $t < T$ that $\zeta_{j,r,i}^{(t)} \leq 0.5\alpha$. Then we have

$$\begin{aligned}
\zeta_{j,r,i}^{(T)} = \zeta_{j,r,i}^{(T_{j,r,i})} &- \underbrace{\frac{\eta}{n} \ell_i'^{(T_{j,r,i})} \cdot \sigma' \left( \left\langle \widetilde{\mathbf{w}}_{j,r}^{(T_{j,r,i})}, \boldsymbol{\xi}_i \right\rangle \right) \mathbb{I}(y_i = j) \left\| \widetilde{\boldsymbol{\xi}}_{j,r,i} \right\|_2^2}_{I_1} \\
&- \underbrace{\sum_{T_{j,r,i} < t < T} \frac{\eta}{n} \ell_{j,i}'^{(T_{j,r,i})} \sigma' \left( \left\langle \widetilde{\mathbf{w}}_{j,r}^{(t)}, \boldsymbol{\xi}_i \right\rangle \right) \mathbb{I}(y_i = j) \left\| \widetilde{\boldsymbol{\xi}}_{j,r,i} \right\|_2^2}_{I_2}.
\end{aligned}$$

We bound $I_1, I_2$ separately. We first bound $I_1$ as follows.

$$|I_1| \leq q \frac{\eta}{n} \left( \zeta_{j,r,i}^{(T_{j,r,i})} + 0.5\beta + 3Cn\alpha \sqrt{\frac{\log d}{pd}} \right)^{q-1} C_2 \sigma_n^2 pd \leq q 2^q n^{-1} \eta \alpha^{q-1} C_2 \sigma_n^2 pd \leq 0.25\alpha,$$

where the first inequality follows from Lemma D.13, the second inequality follows because $\beta \leq 0.1\alpha$ and $3Cn\alpha\sqrt{\frac{\log d}{pd}} \leq 0.1\alpha$, and the last inequality follows because $\eta \leq n/(q2^{q+2}\alpha^{q-2}\sigma_n^2 d)$.

For $T_{j,r,i} < t < T$, by Lemma D.11, we have that $\left\langle \widetilde{\mathbf{w}}_{j,r}^{(t)}, \boldsymbol{\xi}_i \right\rangle \geq 0.5\alpha - 0.5\beta - 3Cn\alpha\sqrt{\frac{\log d}{pd}} \geq 0.25\alpha$ and $\left\langle \widetilde{\mathbf{w}}_{j,r}^{(t)}, \boldsymbol{\xi}_i \right\rangle \leq \alpha + 0.5\beta + 3Cn\alpha\sqrt{\frac{\log d}{pd}} \leq 2\alpha$.

Now we bound $I_2$ as follows

$$
\begin{aligned}
|I_2| &\leq \sum_{T_{j,r,i} < t < T} \frac{\eta}{n} K e \exp\left\{-F_j(X)\right\} \sigma'\left(\left\langle \widetilde{\mathbf{w}}_{j,r}^{(t)}, \boldsymbol{\xi}_i \right\rangle\right) \mathbb{I}(y_i = j) \left\|\widetilde{\boldsymbol{\xi}}_{j,r,i}\right\|_2^2 \\
&\leq \sum_{T_{j,r,i} < t < T} \frac{\eta}{n} K e \exp\left\{-\sigma\left(\left\langle \widetilde{\mathbf{w}}_{j,r}^{(t)}, \boldsymbol{\xi}_i \right\rangle\right)\right\} \sigma'\left(\left\langle \widetilde{\mathbf{w}}_{j,r}^{(t)}, \boldsymbol{\xi}_i \right\rangle\right) \mathbb{I}(y_i = j) \left\|\widetilde{\boldsymbol{\xi}}_{j,r,i}\right\|_2^2 \\
&\leq \frac{qKe\eta 2^{q-1} T^\star}{n} \exp(-\alpha^q/4^q) \alpha^{q-1} \sigma_n^2 pd \\
&\leq 0.25 T^\star \exp(-\alpha^q/4^q) \alpha^{q-2} \alpha \\
&\leq 0.25 \alpha,
\end{aligned}
$$

where the first inequality follows from Equation equation 2, the second inequality follows because $F_j(X) \geq \sigma\left(\left\langle \widetilde{\mathbf{w}}_{j,r}^{(t)}, \boldsymbol{\xi}_i \right\rangle\right)$, the fourth inequality follows by choosing $\eta \leq n/(qKe2^{q+1}\sigma_n^2 d)$, and the last inequality follows by choosing $\alpha = \Theta(\log^{1/q}(T^\star))$.

Plugging the bounds on $I_1, I_2$ finishes the proof for $\zeta_{j,r,i}^{(T)}$.

To prove $\gamma_{j,r,j}^{(t)} \leq \alpha$, we pick $\eta \leq 1/(qe2^{q+2}\mu^2)$ and the rest of the proof is similar. $\qquad\square$

Lemma D.14 and Lemma D.15 imply Proposition D.10 holds for all $t \leq T$.

**Induction Ends** $\qquad\square$

### D.3 Feature Growing Phase

In this subsection, we first present a supporting lemma, and then provide our main result of Lemma D.17, which shows that the signal strength is relatively unaffected by pruning while the noise level is reduced by a factor of $\sqrt{p}$.

During the feature growing phase of training, the output of $F_j(X) = O(1)$ for all $j \in [K]$. Therefore, $\text{logit}_i(F, X) = O(\frac{1}{K})$ and $1 - \text{logit}_i(F, X) = \Theta(1)$ until $\left\langle \widetilde{\mathbf{w}}_{j,r}^{(t)}, \boldsymbol{\mu}_j \right\rangle$ reaches $m^{-1/q}$.

**Lemma D.16.** *Under the same assumption as Theorem D.1, for $T = \frac{n\eta^{-1}C_4\sigma_0^{2-q}(\sigma_n\sqrt{pd})^{-q}}{C_3(2C_1)^{q-1}[\log d]^{(q-1)/2}}$, the following results hold:*

- $|\zeta_{j,r,i}^{(t)}| = O(\sigma_0\sigma_n\sqrt{pd})$ *for all $j \in [K]$, $r \in [m]$, $i \in [n]$ and $t \leq T$.*

- $|\omega_{j,r,i}^{(t)}| = O(\sigma_0\sigma_n\sqrt{pd})$ *for all $j \in [K]$, $r \in [m]$, $i \in [n]$ and $t \leq T$.*

*Proof.* Define $\Psi^{(t)} = \max_{j,r,i}\{\zeta_{j,r,i}^{(t)}, |\omega_{j,r,i}^{(t)}|\}$. Then we have

$\Psi^{(t+1)}$

$$
\leq \Psi^{(t)} + \max_{j,r,i}\left\{\frac{\eta}{n}|\ell_{j,i}'^{(t)}| \cdot \sigma'\left(\left\langle \widetilde{\mathbf{w}}_{j,r}^{(0)}, \boldsymbol{\xi}_i \right\rangle + \sum_{k=1}^{K} \gamma_{j,r,k}^{(t)} \frac{\langle \boldsymbol{\mu}_k, \boldsymbol{\xi}_i \rangle}{\|\boldsymbol{\mu}_k\|_2^2} + \sum_{i'=1}^{n} \Psi^{(t)} \frac{\left\langle \widetilde{\boldsymbol{\xi}}_{j,r,i'}, \boldsymbol{\xi}_i \right\rangle}{\left\|\widetilde{\boldsymbol{\xi}}_{j,r,i'}\right\|_2^2}\right) \left\|\widetilde{\boldsymbol{\xi}}_{j,r,i}\right\|_2^2\right\}
$$

$$
\leq \Psi^{(t)} + \frac{\eta}{n}q\left(O(\sqrt{\log d}\sigma_0\sigma_n\sqrt{pd}) + K\log^{1/q}T^\star \frac{\mu\sigma_n\sqrt{\log d}}{\mu^2} + \frac{O(\sigma_n^2 pd) + nO(\sigma_n^2\sqrt{pd\log d})}{\Theta(\sigma_n^2 pd)}\Psi^{(t)}\right)^{q-1} O(\sigma_n^2 pd)
$$

$$
\leq \Psi^{(t)} + \frac{\eta}{n}\left(O(\sqrt{\log d}\sigma_0\sigma_n\sqrt{pd}) + O(\Psi^{(t)})\right)^{q-1} O(\sigma_n^2 pd),
$$

where the second inequality follows by $|\ell_{j,i}'^{(t)}|$ and applying the bounds from Lemma D.5, and the last inequality follows by choosing $\frac{2K\log T^\star}{\sigma_n d\sqrt{p}} = \widetilde{O}(1/\sqrt{d}) \ll \sigma_0$. Let $C_1, C_2, C_3$ be the constants for the upper bound to hold in the big O notation. For any $T = \frac{n\eta^{-1}C_4\sigma_0^{2-q}(\sigma_n\sqrt{pd})^{-q}}{C_3(2C_1)^{q-1}[\log d]^{(q-1)/2}} =$

$\Theta(\frac{n\eta^{-1}\sigma_0^{2-q}(\sigma_n\sqrt{pd})^{-q}}{[\log d]^{(q-1)/2}})$, we use induction to show that

$$\Psi^{(t)} \leq C_4\sigma_0\sigma_n\sqrt{pd}, \ \forall t \in [T]. \tag{3}$$

Suppose that Equation equation 3 holds for $t \in [T']$ for $T' \leq T - 1$. Then

$$
\begin{aligned}
\Psi^{(T'+1)} &\leq \Psi^{(T')} + \frac{\eta}{n}\left(C_1\sqrt{\log d}\sigma_0\sigma_n\sqrt{pd} + C_2C_4\sigma_0\sigma_n\sqrt{pd}\right)^{q-1}C_3\sigma_n^2 pd \\
&\leq \Psi^{(T')} + \frac{\eta}{n}\left(2C_1\sqrt{\log d}\sigma_0\sigma_n\sqrt{pd}\right)^{q-1}C_3\sigma_n^2 pd \\
&\leq (T'+1)\frac{\eta}{n}\left(2C_1\sqrt{\log d}\sigma_0\sigma_n\sqrt{pd}\right)^{q-1}C_3\sigma_n^2 pd \\
&\leq T\frac{\eta}{n}\left(2C_1\sqrt{\log d}\sigma_0\sigma_n\sqrt{pd}\right)^{q-1}C_3\sigma_n^2 pd \\
&\leq C_4\sigma_0\sigma_n\sqrt{pd},
\end{aligned}
$$

where the last inequality follows by picking $T = \frac{n\eta^{-1}C_4\sigma_0^{2-q}(\sigma_n\sqrt{pd})^{-q}}{C_3(2C_1)^{q-1}[\log d]^{(q-1)/2}} = \Theta(\frac{n\eta^{-1}\sigma_0^{2-q}(\sigma_n\sqrt{pd})^{-q}}{[\log d]^{(q-1)/2}})$. Therefore, by induction, we have $\Psi^{(t)} \leq C_4\sigma_0\sigma_n\sqrt{pd}$ for all $t \in [T]$. $\quad\square$

**Lemma D.17** (Formal Restatement of Lemma 3.3). *Under the same assumption as Theorem D.1, there exists time $T_1 = \frac{\log 2m^{-1/q}}{\log(1+\Theta(\frac{\eta}{K})\mu^q\sigma_0^{q-2})} = O(K\eta^{-1}\sigma_0^{2-q}\mu^{-q}\log 2m^{-1/q})$ such that*

1. $\max_r \gamma_{j,r,j}^{(T_1)} \geq m^{-1/q}$ *for $j \in [K]$.*

2. $|\zeta_{j,r,i}^{(t)}|, |\omega_{j,r,i}^{(t)}| \leq O(\sigma_0\sigma_n\sqrt{pd})$ *for all $j \in [K], r \in [m], i \in [n]$ and $t \leq T_1$.*

3. $|\gamma_{j,r,k}^{(t)}| \leq O(\sigma_0\mu\,\mathrm{poly}\log d)$ *for all $j, k \in [K], \ j \neq k, \ r \in [m]$ and $t \leq T_1$.*

*Proof.* Consider a fixed class $j \in [K]$. Denote $T_1$ to be the last time for $t \in \left[0, \frac{n\eta^{-1}C_4\sigma_0^{2-q}(\sigma_n\sqrt{pd})^{-q}}{C_3(2C_1)^{q-1}[\log d]^{(q-1)/2}}\right]$ satisfying $\max_r \gamma_{j,r}^{(t)} \leq m^{-1/q}$. Then for $t \leq T_1$, $\max_{j,r,i} \zeta_{j,r,i}^{(t)}, |\omega_{j,r,i}^{(t)}| \leq O(\sigma_0\sigma_p\sqrt{pd}) \leq O(m^{-1/q})$ and $\max_{j,r} \gamma_{j,r,j}^{(t)}$. Thus, by Lemma D.13, we obtain that $F_j(\widetilde{\mathbf{W}}^{(t)}, \mathbf{x}_i) \leq O(1), \forall y_i = j$. Thus, $\ell_{j,i}^{'(t)} = \Theta(1)$. For $j \in \mathcal{S}_{\mathrm{signal}}^j$, we have

$\gamma_{j,r,j}^{(t+1)}$

$$
= \gamma_{j,r,j}^{(t)} - \frac{\eta}{n}\sum_{i=1}^n \ell_{j,i}^{'(t)} \cdot \sigma'\left(\left\langle\widetilde{\mathbf{w}}_{j,r}^{(0)}, \boldsymbol{\mu}_j\right\rangle + \gamma_{j,r,j}^{(t)} + \sum_{i'=1}^n \zeta_{j,r,i}^{(t)}\frac{\left\langle\widetilde{\boldsymbol{\xi}}_{j,r,i}, \boldsymbol{\mu}_j\right\rangle}{\left\|\widetilde{\boldsymbol{\xi}}_{j,r,i}\right\|_2^2} + \sum_{i'=1}^n \omega_{j,r,i}^{(t)}\frac{\left\langle\widetilde{\boldsymbol{\xi}}_{j,r,i}, \boldsymbol{\mu}_j\right\rangle}{\left\|\widetilde{\boldsymbol{\xi}}_{j,r,i}\right\|_2^2}\right)\|\boldsymbol{\mu}_j\|_2^2\,\mathbb{I}(y_i = j)
$$

$$
\geq \gamma_{j,r,j}^{(t)} - \frac{\eta}{n}\sum_{i=1}^n \ell_{j,i}^{'(t)}\sigma'\left(\left\langle\widetilde{\mathbf{w}}_{j,r}^{(0)}, \boldsymbol{\mu}_j\right\rangle + \gamma_{j,r,j}^{(t)} - O(n\sigma_0\sigma_n pd\frac{\sigma_n\mu\sqrt{\log d}}{\sigma_n^2 pd})\right)\mathbb{I}(y_i = j).
$$

Let $\widehat{\gamma}_{j,r,j}^{(t)} = \gamma_{j,r,j}^{(t)} + \left\langle\widehat{\mathbf{w}}_{j,r}^{(0)}, \boldsymbol{\mu}_j\right\rangle - O(n\sigma_0\sigma_n\sqrt{pd}\frac{\sigma_n\mu\sqrt{\log d}}{\sigma_n^2 pd})$ and $A^{(t)} = \max_r \widehat{\gamma}_{j,r,j}^{(t)}$. Note that by our choice of $\mu$, we have $\frac{n\mu\sqrt{\log d}}{\sigma_n pd} = o(1)$. Since $\max_r\left\langle\widetilde{\mathbf{w}}_{j,r}^{(0)}, \boldsymbol{\mu}_j\right\rangle \geq \Omega(\sigma_0\mu)$ by Lemma D.7, $\max_r\left\langle\widetilde{\mathbf{w}}_{j,r}^{(0)}, \boldsymbol{\mu}_j\right\rangle \geq \Omega(\sigma_0\mu) - O(n\sigma_0\sigma_n pd\frac{\sigma_n\mu\sqrt{\log d}}{\sigma_n^2 pd}) = \Omega(\sigma_0\mu)$. Then we have

$$
\begin{aligned}
A^{(t+1)} &\geq A^{(t)} - \frac{\eta}{n}\sum_{i=1}^n \ell_{j,i}^{'(t)}\sigma'(A^{(t)})\mu^2\mathbb{I}(y_i = j) \\
&\geq A^{(t)} + \Theta(\frac{\eta}{K})\mu^2[A^{(t)}]^{q-1} \\
&\geq (1 + \Theta(\frac{\eta}{K}\mu^2[A^{(t)}]^{q-2}))A^{(t)} \\
&\geq (1 + \Theta(\frac{\eta}{K}\mu^q\sigma_0^{q-2}))A^{(t)}.
\end{aligned}
$$

Therefore, the sequence $A^{(t)}$ will exponentially grow and will reach $2m^{-1/q}$ within $\frac{\log 2m^{-1/q}}{\log(1+\Theta(\frac{\eta}{K})\mu^q \sigma_0^{q-2})} = O(K\eta^{-1}\sigma_0^{2-q}\mu^{-q}\log 2m^{-1/q}) \leq \Theta(\frac{n\eta^{-1}\sigma_0^{2-q}(\sigma_n\sqrt{pd})^{-q}}{[\log d]^{(q-1)/2}})$. Thus, $\max_r \gamma_{j,r}^{(t)} \geq A^{(t)} - \max_{j,r}|\langle \widetilde{\mathbf{w}}_{j,r}^{(0)}, \boldsymbol{\mu}_j\rangle| \geq 2m^{-1/q} - O(\sigma_0\mu) \geq 2 - m^{-1/q} = m^{-1/q}$.

Now we prove that under the same assumption as Theorem D.1, for $T = O(K\eta^{-1}\sigma_0^{2-q}\mu^{-q})$, we have $|\gamma_{j,r,k}^{(t)}| \leq O(\sigma_0\mu\,\text{poly}\log d)$ for all $r \in [m]$, $j,k \in [K]$, $j \neq k$ and $t \leq T$.

We show that there exists a time $T' \geq T$ such that for all $t \leq T'$, $\max_{j,r,k}|\gamma_{j,r,k}^{(t)}| \leq O(\sigma_0\mu\,\text{poly}\log d)$. Let $T' = O(K^2\eta^{-1}\sigma_0^{2-q}\mu^{-q}\log d)$.

Define $\Phi^{(t)} = \max_{r\in[m],\,j,k\in[K],\,j\neq k}\{|\gamma_{j,r,k}^{(t)}|\}$. Since we assume $T \leq \Theta(\frac{n\eta^{-1}\sigma_0^{2-q}(\sigma_n\sqrt{pd})^{-q}}{[\log d]^{(q-1)/2}})$, by Lemma D.16, we have $\zeta_{j,r,i}^{(t)}, |\omega_{j,r,i}^{(t)}| \leq O(\sigma_0\sigma_n\sqrt{pd})$.

$\Phi^{(t+1)}$

$$\leq \Phi^{(t)} + \max_{j,r,k,i}\left\{\frac{\eta}{n}\sum_{i=1}^n \mathbb{I}(y_i=k)|\ell_{j,i}'^{(t)}|\sigma'\left(\langle\widetilde{\mathbf{w}}_{j,r}^{(0)},\boldsymbol{\mu}_k\rangle + \sum_{i'=1}^n \zeta_{j,r,i'}^{(t)}\frac{\langle\widetilde{\boldsymbol{\xi}}_{j,r,i'},\boldsymbol{\mu}_k\rangle}{\left\|\widetilde{\boldsymbol{\xi}}_{j,r,i'}\right\|_2^2} + \sum_{i'=1}^n \omega_{j,r,i'}^{(t)}\frac{\langle\widetilde{\boldsymbol{\xi}}_{j,r,i'},\boldsymbol{\mu}_k\rangle}{\left\|\widetilde{\boldsymbol{\xi}}_{j,r,i'}\right\|_2^2}\right)\mu^2\right\}$$

$$\leq \Phi^{(t)} + \frac{\eta}{K}\frac{1}{K}q\left(O(\sigma_0\mu\sqrt{\log d}) + nO(\sigma_0\sigma_n\sqrt{pd})\frac{\sigma_n\mu\sqrt{\log d}}{\sigma_n^2 pd})\right)^{q-1}\mu^2$$

$$\leq \Phi^{(t)} + \frac{q\eta}{K^2}\left(O(\sigma_0\mu\sqrt{\log d})\right)^{q-1}\mu^2,$$

where the first inequality follows because $\gamma_{j,r,k}^{(t)} < 0$, the second inequality follows because there are $\Theta(n/K)$ samples from a given class $k$ and $|\ell_{j,i}'^{(t)}| = \Theta(\frac{1}{K})$, and the last inequality follows because $\mu = \sigma_n\sqrt{d}\log d$. Now, let $C$ be the constant such that the above holds with big O. Then, we use induction to show that $\Phi^{(t)} \leq C_2\sigma_0\mu$ for all $t \leq T$. We proceed as follows.

$$\Phi^{(t+1)} \leq \Phi^{(t)} + \frac{q\eta}{K^2}\left(C\sigma_0\mu\sqrt{\log d}\right)^{q-1}\mu^2$$
$$\leq T\frac{q\eta}{K^2}\left(C\sigma_0\mu\sqrt{\log d}\right)^{q-1}\mu^2$$
$$\leq C_2\sigma_0\mu\,\text{poly}\log d,$$

where the last inequality follows by picking $T = \frac{C_2 K^2\eta^{-1}\sigma_0^{2-q}\mu^{-q}\sqrt{\log d}}{C^{q-1}} = O(K^2\eta^{-1}\sigma_0^{2-q}\mu^{-q}\log d)$. $\qquad\square$

## D.4 Converging Phase

In this subsection, we show that gradient descent can drive the training loss toward zero while the signal in the feature is still large. An important intermediate step in our argument is the development of the following gradient upper bound for multi-class cross-entropy loss.

In this phase, we are going to show that

- $\max_r \gamma_{j,r,j}^{(t)} \geq m^{1/q}$ for all $j \in [K]$.

- $\max_{j\neq k, r\in[m]}|\gamma_{j,r,k}^{(t)}| \leq \beta_1$ where $\beta_1 = \widetilde{O}(\sigma_0\mu)$.

- $\max_{j,r,i}\{\zeta_{j,r,i}^{(t)}, |\omega_{j,r,i}^{(t)}|\} \leq \beta_2$ where $\beta_2 = O(\sigma_0\sigma_n\sqrt{pd})$

Define $\mathbf{W}^\star$ as follows:

$$\mathbf{w}_{j,r}^\star = \mathbf{w}_{j,r}^{(0)} + \Theta(m\log(1/\epsilon))\frac{\boldsymbol{\mu}_j}{\mu^2}.$$

**Lemma D.18.** *Based on the result from the feature growing phase,* $\left\|\widetilde{\mathbf{W}}^{(T_1)} - \widetilde{\mathbf{W}}^\star\right\|_F^2 \leq O(Km^3 \log^2(1/\epsilon)\mu^{-2})$.

*Proof.* We first compute

$$\left\|\widetilde{\mathbf{W}}^{(T_1)} - \widetilde{\mathbf{W}}^{(0)}\right\|_F^2$$

$$= \sum_{j=1}^K \sum_{r=1}^m \left\| \gamma_{j,r,j}^{(T_1)} \frac{\boldsymbol{\mu}_j \odot \mathbf{m}_{j,r}}{\mu^2} + \sum_{k\neq j} \gamma_{j,r,k}^{(T_1)} \frac{\boldsymbol{\mu}_k \odot \mathbf{m}_{j,r}}{\mu^2} + \sum_i \zeta_{j,r,i}^{(T_1)} \frac{\widetilde{\boldsymbol{\xi}}_{j,r,i}}{\left\|\widetilde{\boldsymbol{\xi}}_{j,r,i}\right\|_2^2} + \sum_i \omega_{j,r,i}^{(T_1)} \frac{\widetilde{\boldsymbol{\xi}}_{j,r,i}}{\left\|\widetilde{\boldsymbol{\xi}}_{j,r,i}\right\|_2^2} \right\|_2^2$$

$$\leq \sum_j \sum_r \left( \gamma_{j,r,j}^{(T_1)} \frac{1}{\mu} + \sum_{k\neq j} \gamma_{j,r,k}^{(T_1)} \frac{1}{\mu} + \sum_i \zeta_{j,r,i}^{(T_1)} \frac{1}{\left\|\widetilde{\boldsymbol{\xi}}_{j,r,i}\right\|_2} + \sum_i \omega_{j,r,i}^{(T_1)} \frac{1}{\left\|\widetilde{\boldsymbol{\xi}}_{j,r,i}\right\|_2} \right)^2$$

$$\leq \sum_j \sum_r \left( \widetilde{O}(\frac{1}{\mu}) + K\widetilde{O}(\sigma_0) + n\widetilde{O}(\sigma_0) \right)^2$$

$$\leq \sum_j \sum_r \widetilde{O}(\frac{1}{\mu^2})$$

$$= \widetilde{O}(Km \frac{1}{\mu^2}),$$

where the first inequality follows from triangle inequality, the second inequality follows from Lemma D.17, and the last inequality follows from our choice of $\sigma_0$. On the other hand,

$$\left\|\widetilde{\mathbf{W}}^{(0)} - \widetilde{\mathbf{W}}^\star\right\|_F^2 = \sum_{j,r} m^2 \log^2(1\epsilon) \frac{1}{\mu^2} = O(Km^3 \log^2(1/\epsilon) \frac{1}{\mu^2}).$$

Thus, we obtain

$$\left\|\widetilde{\mathbf{W}}^{(T_1)} - \widetilde{\mathbf{W}}^\star\right\|_F^2 \leq 4\left\|\widetilde{\mathbf{W}}^{(T_1)} - \widetilde{\mathbf{W}}^{(0)}\right\|_F^2 + 4\left\|\widetilde{\mathbf{W}}^{(0)} - \widetilde{\mathbf{W}}^\star\right\|_F^2 \leq O(Km^3 \log^2(1/\epsilon) \frac{1}{\mu^2}).$$

$\square$

**Lemma D.19** (Gradient Upper Bound). *Under Condition 2.2, for $t \leq T^\star$, there exists constant $C = O(Km^{2/q} \max\{\mu^2, \sigma_n^2 pd\})$ such that*

$$\left\|\nabla L_S(\widetilde{\mathbf{W}}^{(t)}) \odot \mathbf{M}\right\|_F^2 \leq C L_S(\widetilde{\mathbf{W}}^{(t)}).$$

*Proof.* We need to prove that $|\ell_{y_i,i}'^{(t)}| \left\|\nabla F(\widetilde{\mathbf{W}}^{(t)}, \mathbf{x}_i) \odot \mathbf{M}\right\|_F^2 \leq C$. Assume $y_i \neq j$. Then we obtain

$$\left\|\nabla F_j(\widetilde{\mathbf{W}}_j, \mathbf{x}_i) \odot \mathbf{M}\right\|_F \leq \sum_r \left\|\sigma'\left(\left\langle\widetilde{\mathbf{w}}_{j,r}^{(t)}, \boldsymbol{\mu}_{y_i}\right\rangle\right) \boldsymbol{\mu}_{y_i} + \sigma'\left(\left\langle\widetilde{\mathbf{w}}_{j,r}^{(t)}, \boldsymbol{\xi}_i\right\rangle\right) \widetilde{\boldsymbol{\xi}}_i \right\|_2$$

$$\leq \sum_r \sigma'\left(\left\langle\widetilde{\mathbf{w}}_{j,r}^{(t)}, \boldsymbol{\mu}_{y_i}\right\rangle\right) \|\boldsymbol{\mu}_{y_i}\|_2 + \sigma'\left(\left\langle\widetilde{\mathbf{w}}_{j,r}^{(t)}, \boldsymbol{\xi}_i\right\rangle\right) \left\|\widetilde{\boldsymbol{\xi}}_i\right\|_2$$

$$\leq m^{1/q} \left[F_j(\widetilde{\mathbf{W}}_j, \mathbf{x}_i)\right]^{(q-1)/q} \max\{\mu, C\sigma_n\sqrt{pd}\}$$

$$\leq m^{1/q} \max\{\mu, C\sigma_n\sqrt{pd}\},$$

where the first and second inequality follow from triangle inequality, the third inequality follows from Hölder's inequality, and the last inequality follows from Lemma D.12. Similarly, on the other hand, if $y_i = j$, then

$$\left\|\nabla F_{y_i}(\widetilde{\mathbf{W}}) \odot \mathbf{M}\right\|_F \leq m^{1/q} \left[F_{y_i}(\widetilde{\mathbf{W}}_{y_i}, \mathbf{x}_i)\right]^{(q-1)/q} \max\{\mu, C\sigma_n\sqrt{pd}\}.$$

Therefore,

$$\sum_{j \neq y_i} |\ell_{j,i}'^{(t)}| \left\| \nabla F_j(\widetilde{\mathbf{W}}_j, \mathbf{x}_i) \odot \mathbf{M}_j \right\|_F^2 \leq \sum_{j \neq y_i} |\ell_{j,i}'^{(t)}| m^{2/q} O(\max\{\mu^2, \sigma_n^2 pd\})$$
$$= |\ell_{y_i,i}'^{(t)}| m^{2/q} O(\max\{\mu^2, \sigma_n^2 pd\})$$
$$\leq K e \exp\{-F_{y_i}(\mathbf{x}_i)\} m^{2/q} O(\max\{\mu^2, \sigma_n^2 pd\}),$$

and

$$|\ell_{y_i,i}'^{(t)}| \left\| \nabla F_{y_i}(\widetilde{\mathbf{W}}_{y_i}, \mathbf{x}_i) \odot \mathbf{M}_{y_i} \right\|_F^2$$
$$\leq K e \exp\{-F_{y_i}(\mathbf{x}_i)\} m^{2/q} \left[ F_{y_i}(\widetilde{\mathbf{W}}_{y_i}, \mathbf{x}_i) \right]^{2(q-1)/q} O(\max\{\mu^2, \sigma_n^2 pd\},$$

where the inequality follows from Equation equation 2. Thus,

$$\sum_{j=1}^{K} |\ell_{j,i}'^{(t)}|^2 \left\| \nabla F_j(\widetilde{\mathbf{W}}_j, \mathbf{x}_i) \odot \mathbf{M}_j \right\|_F^2$$
$$\leq |\ell_{y_i,i}'^{(t)}| \sum_{j=1}^{K} |\ell_{j,i}'^{(t)}| \left\| \nabla F_j(\widetilde{\mathbf{W}}_j, \mathbf{x}_i) \odot \mathbf{M}_j \right\|_F^2$$
$$\leq |\ell_{y_i,i}'^{(t)}| K e \exp\{-F_{y_i}(\mathbf{x}_i)\} m^{2/q} O(\max\{\mu^2, \sigma_n^2 pd\} \left( \left[ F_{y_i}(\widetilde{\mathbf{W}}_{y_i}, \mathbf{x}_i) \right]^{(q-1)/q} + 1 \right)$$
$$\leq |\ell_{y_i,i}'^{(t)}| O(K m^{2/q} \max\{\mu^2, \sigma_n^2 pd\}), \tag{4}$$

where the first inequality follows because $|\ell_{j,i}'^{(t)}| \leq |\ell_{y_i,i}'^{(t)}|$, and the last inequality uses the fact that $\exp\{-x\}(1 + x^{(q-1)/q}) = O(1)$ for all $x \geq 0$.

The gradient norm can be bounded by

$$\left\| \nabla L_S(\widetilde{\mathbf{W}}^{(t)}) \right\|_F^2 \leq \left( \frac{1}{n} \sum_{i=1}^{n} \left\| \nabla L(\widetilde{\mathbf{W}}^{(t)}, \mathbf{x}_i) \right\|_F \right)^2$$
$$= \left( \frac{1}{n} \sum_{i=1}^{n} \sqrt{\sum_{j=1}^{K} |\ell_{j,i}'^{(t)}|^2 \left\| \nabla F_j(\widetilde{\mathbf{W}}_j^{(t)}, \mathbf{x}_i) \right\|_F^2} \right)^2$$
$$\leq \left( \frac{1}{n} \sum_{i=1}^{n} |\ell_{y_i,i}'^{(t)}| \left\| \nabla F(\widetilde{\mathbf{W}}_j^{(t)}, \mathbf{x}_i) \right\|_F \right)^2$$
$$\leq \left( \frac{1}{n} \sum_{i=1}^{n} \sqrt{|\ell_{y_i,i}'^{(t)}| O(K m^{2/q} \max\{\mu^2, \sigma_n^2 d\})} \right)^2$$
$$\leq O(K m^{2/q} \max\{\mu^2, \sigma_n^2 d\}) \frac{1}{n} \sum_{i=1}^{n} |\ell_{y_i,i}'^{(t)}|$$
$$\leq O(K m^{2/q} \max\{\mu^2, \sigma_n^2 d\}) L_S(\widetilde{\mathbf{W}}^{(t)}),$$

where the first inequality uses triangle inequality, the second inequality follows because $|\ell_{j,i}'^{(t)}| \leq |\ell_{y_i,i}'^{(t)}|$, the third inequality uses the bound equation 4, the fourth inequality uses Jensen's inequality and the last inequality follows because $|\ell_{y_i,i}'^{(t)}| \leq \ell_i^{(t)}$. $\qquad \square$

**Lemma D.20.** *For $T_1 \leq t \leq T^\star$, we have for all $j \neq y_i$,*

$$\left\langle \nabla F_{y_i}(\widetilde{\mathbf{W}}_{y_i}^{(t)}, \mathbf{x}_i), \widetilde{\mathbf{W}}_{y_i}^\star \right\rangle - \left\langle \nabla F_j(\widetilde{\mathbf{W}}_j^{(t)}, \mathbf{x}_i), \widetilde{\mathbf{W}}_j^\star \right\rangle \geq q \log \frac{2qK}{\epsilon}.$$

*Proof of Lemma D.20.* The proof of this lemma depends on the next two lemmas.

**Lemma D.21.** *For $T_1 \leq t \leq T^\star$ and $j = y_i$, we have $\left\langle \nabla F_j(\widetilde{\mathbf{W}}_j^{(t)}, \mathbf{x}_i), \widetilde{\mathbf{W}}_j^\star \right\rangle \geq \Theta(m^{1/q} \log(1/\epsilon))$.*

*Proof.* By Lemma D.17, we have

$$
\max_r \left\{ \left\langle \widetilde{\mathbf{w}}_{j,r}^{(t)}, \boldsymbol{\mu}_j \right\rangle \right\} = \max_r \left\{ \left\langle \widetilde{\mathbf{w}}_{j,r}^{(0)}, \boldsymbol{\mu}_j \right\rangle + \gamma_{j,r,j}^{(t)} + \sum_{i=1}^n \zeta_{j,r,i}^{(t)} \frac{\left\langle \widetilde{\boldsymbol{\xi}}_{j,r,i}, \boldsymbol{\mu}_j \right\rangle}{\left\| \widetilde{\boldsymbol{\xi}}_{j,r,i} \right\|_2^2} + \sum_{i=1}^n \omega_{j,r,i}^{(t)} \frac{\left\langle \widetilde{\boldsymbol{\xi}}_{j,r,i}, \boldsymbol{\mu}_j \right\rangle}{\left\| \widetilde{\boldsymbol{\xi}}_{j,r,i} \right\|_2^2} \right\}
$$

$$
\geq m^{-1/q} - O(\sigma_0 \mu \sqrt{\log d}) - O(n \sigma_0 \sigma_n \sqrt{d} \frac{\mu \sqrt{\log d}}{\sigma_n p d})
$$

$$
\geq \Theta(m^{-1/q}),
$$

where the last inequality follows by picking $\sigma_0 \leq O(m^{-1} n^{-1} \mu^{-1} (\log d)^{-1/2})$. On the other hand,

$$
\left| \left\langle \widetilde{\mathbf{w}}_{j,r}^{(t)}, \boldsymbol{\xi}_i \right\rangle \right| \leq \left| \left\langle \widetilde{\mathbf{w}}_{j,r}^{(0)}, \boldsymbol{\xi}_i \right\rangle \right| + |\omega_{j,r,i}^{(t)}| + |\zeta_{j,r,i}^{(t)}| + O(n \sqrt{\frac{\log d}{pd}} \alpha) + O(n \frac{\mu \sqrt{\log d}}{\sigma_n p d} \alpha) \leq O(1),
$$

(5)

where the first inequality follows from Lemma D.11 and the second inequality follows from Equation (1) and Proposition D.10. Therefore,

$$
\left\langle \nabla F_j(\widetilde{\mathbf{W}}_j^{(t)}, \mathbf{x}_i), \widetilde{\mathbf{W}}_j^\star \right\rangle
$$

$$
= \sum_r \sigma' \left( \left\langle \widetilde{\mathbf{w}}_{j,r}^{(t)}, \boldsymbol{\mu}_j \right\rangle \right) \left\langle \boldsymbol{\mu}_j, \widetilde{\mathbf{w}}_{j,r}^\star \right\rangle + \sum_r \sigma' \left( \left\langle \widetilde{\mathbf{w}}_{j,r}^{(t)}, \boldsymbol{\xi}_i \right\rangle \right) \left\langle \boldsymbol{\xi}_i, \widetilde{\mathbf{w}}_{j,r}^\star \right\rangle
$$

$$
\geq \sum_r \sigma' \left( \left\langle \widetilde{\mathbf{w}}_{j,r}^{(t)}, \boldsymbol{\mu}_j \right\rangle \right) \Theta(m \log(1/\epsilon)) - \sum_r O(\sigma_0 \sigma_n \sqrt{pd \log d} + \frac{\sigma_n \sqrt{\log d}}{\mu} m \log(1/\epsilon))
$$

$$
\geq \Theta(m^{1/q} \log(1/\epsilon)) - O(m \sigma_0 \sigma_n \sqrt{pd \log d} + \frac{\sigma_n \sqrt{\log d}}{\mu} m^2 \log(1/\epsilon))
$$

$$
\geq \Theta(m^{1/q} \log(1/\epsilon)),
$$

where the last inequality follows because $m \sigma_0 \sigma_n \sqrt{pd \log d} = o(1)$ and $\frac{\sigma_n \sqrt{\log d}}{\mu} m^2 = o(1)$ by our choices of $\mu, \sigma_0$. $\square$

**Lemma D.22.** *For $T_1 \leq t \leq T$ and $j \neq y_i$, we have $\left\langle \nabla F_j(\widetilde{\mathbf{W}}_j^{(t)}, \mathbf{x}_i), \widetilde{\mathbf{W}}_j^\star \right\rangle \leq O(1)$.*

*Proof.* First we have

$$
\left\langle \widetilde{\mathbf{w}}_{j,r}^{(t)}, \boldsymbol{\mu}_{y_i} \right\rangle = \left\langle \widetilde{\mathbf{w}}_{j,r}^{(0)}, \boldsymbol{\mu}_{y_i} \right\rangle + \gamma_{j,r,y_i}^{(t)} + \sum_{i=1}^n \zeta_{j,r,i}^{(t)} \frac{\left\langle \widetilde{\boldsymbol{\xi}}_{j,r,i}, \boldsymbol{\mu}_j \right\rangle}{\left\| \widetilde{\boldsymbol{\xi}}_{j,r,i} \right\|_2^2} + \sum_{i=1}^n \omega_{j,r,i}^{(t)} \frac{\left\langle \widetilde{\boldsymbol{\xi}}_{j,r,i}, \boldsymbol{\mu}_j \right\rangle}{\left\| \widetilde{\boldsymbol{\xi}}_{j,r,i} \right\|_2^2}
$$

$$
\leq O(\sigma_0 \mu \sqrt{\log d} + \sigma_0 \mu \operatorname{poly} \log d + n \sigma_0 \sigma_n \sqrt{pd} \frac{\sigma_n \mu \sqrt{\log d}}{\sigma_n^2 p d})
$$

$$
\leq O(1),
$$

(6)

where the first inequality follows from Lemma D.7, Lemma D.5 and Lemma D.17, and the last inequality follows from our choices of $\sigma_0, \mu$. Then, we have

$$
\left\langle \nabla F_j(\widetilde{\mathbf{W}}_j^{(t)}, \mathbf{x}_i), \widetilde{\mathbf{W}}_j^\star \right\rangle
$$

$$
= \sum_r \sigma' \left( \left\langle \widetilde{\mathbf{w}}_{j,r}^{(t)}, \boldsymbol{\mu}_{y_i} \right\rangle \right) \left\langle \boldsymbol{\mu}_{y_i}, \widetilde{\mathbf{w}}_{j,r}^\star \right\rangle + \sum_r \sigma' \left( \left\langle \widetilde{\mathbf{w}}_{j,r}^{(t)}, \boldsymbol{\xi}_i \right\rangle \right) \left\langle \boldsymbol{\xi}_i, \widetilde{\mathbf{w}}_{j,r}^\star \right\rangle
$$

$$
\leq m O(\sigma_0 \mu \sqrt{\log d}) + m O(\sigma_0 \sigma_n \sqrt{d \log d} + m \log(1/\epsilon) \frac{\sigma_n \sqrt{\log d}}{\mu})
$$

$$
\leq O(1),
$$

where the second inequality follows from Equation equation 6 and Equation equation 5, and the last inequality follows from our choices of $\mu, \sigma_0$. □

Applying the lower bound and upper bound from Lemma D.21 and Lemma D.22, we have

$$
\left\langle \nabla F_{y_i}(\widetilde{\mathbf{W}}_{y_i}^{(t)}, \mathbf{x}_i), \widetilde{\mathbf{W}}_{y_i}^{\star} \right\rangle - \left\langle \nabla F_j(\widetilde{\mathbf{W}}_j^{(t)}, \mathbf{x}_i), \widetilde{\mathbf{W}}_j^{\star} \right\rangle
$$
$$
\geq \Theta(m^{1/q} \log(1/\epsilon)) - O(1)
$$
$$
\geq q \log \frac{2qK}{\epsilon}.
$$

□

**Lemma D.23.** *Under the same assumption as Theorem D.1, we have*
$$
\left\| \widetilde{\mathbf{W}}^{(t)} - \widetilde{\mathbf{W}}^{\star} \right\|_F^2 - \left\| \widetilde{\mathbf{W}}^{(t+1)} - \widetilde{\mathbf{W}}^{\star} \right\|_F^2 \geq 5\eta L_S(\widetilde{\mathbf{W}}^{(t)}) - \eta\epsilon.
$$

*Proof.* To simplify our notation, we define $\widehat{F}_j^{(t)}(\mathbf{x}_i) = \left\langle \nabla F_j(\widetilde{\mathbf{W}}_j^{(t)}, \mathbf{x}_i), \widetilde{\mathbf{W}}_j^{\star} \right\rangle$.

We use the fact that the network is $q$-homogeneous.

$$
\left\| \widetilde{\mathbf{W}}^{(t)} - \widetilde{\mathbf{W}}^{\star} \right\|_F^2 - \left\| \widetilde{\mathbf{W}}^{(t+1)} - \widetilde{\mathbf{W}}^{\star} \right\|_F^2
$$
$$
= 2\eta \left\langle \nabla L_S(\widetilde{\mathbf{W}}^{(t)}) \odot \mathbf{M}, \widetilde{\mathbf{W}}^{(t)} - \widetilde{\mathbf{W}}^{\star} \right\rangle - \eta^2 \left\| \nabla L_S(\widetilde{\mathbf{W}}^{(t)}) \odot \mathbf{M} \right\|_F^2
$$
$$
= \frac{2\eta}{n} \sum_{i=1}^{n} \sum_{j=1}^{K} \ell_{j,i}'^{(t)} \left[ qF_j(\widetilde{\mathbf{W}}_j^{(t)}; \mathbf{x}_i, y_i) - \left\langle \nabla F_j(\widetilde{\mathbf{W}}_j^{(t)}, \mathbf{x}_i), \widetilde{\mathbf{W}}_j^{\star} \right\rangle \right] - \eta^2 \left\| \nabla L_S(\widetilde{\mathbf{W}}^{(t)}) \odot \mathbf{M} \right\|_F^2
$$
$$
\geq \frac{2q\eta}{n} \sum_{i=1}^{n} \left[ \log(1 + \sum_{j=1}^{K} e^{F_j - F_{y_i}}) - \log(1 + \sum_{j=1}^{K} e^{(\widehat{F}_j - \widehat{F}_{y_i})/q}) \right] - \eta^2 \left\| \nabla L_S(\widetilde{\mathbf{W}}^{(t)}) \odot \mathbf{M} \right\|_F^2
$$
$$
\geq \frac{2q\eta}{n} \sum_{i=1}^{n} \left[ \ell(\widetilde{\mathbf{W}}^{(t)}; \mathbf{x}_i, y_i) - \log(1 + Ke^{-\log(2qK/\epsilon)}) \right] - \eta^2 \left\| \nabla L_S(\widetilde{\mathbf{W}}^{(t)}) \odot \mathbf{M} \right\|_F^2
$$
$$
\geq \frac{2q\eta}{n} \sum_{i=1}^{n} \left[ \ell(\widetilde{\mathbf{W}}^{(t)}; \mathbf{x}_i, y_i) - \frac{\epsilon}{2q} \right] - \eta^2 \left\| \nabla L_S(\widetilde{\mathbf{W}}^{(t)}) \odot \mathbf{M} \right\|_F^2
$$
$$
\geq C\eta L_S(\widetilde{\mathbf{W}}^{(t)}) - \eta\epsilon,
$$

where the first inequality follows from the convexity of the cross-entropy loss with softmax, the second inequality follows from Lemma D.20, the third inequality follows because $\log(1 + x) \leq x$, and the last inequality follows from Lemma D.19 for some constant $C$. □

**Lemma D.24** (Formal Restatement of Lemma 3.5). *Under the same assumption as Theorem D.1, choose $T_2 = T_1 + \frac{\|\widetilde{\mathbf{W}}^{(T_1)} - \widetilde{\mathbf{W}}^{\star}\|_F^2}{2\eta\epsilon} = T_1 + \widetilde{O}(Km^3 \log^2(1/\epsilon)\mu^{-2})$. Then for any time $t$ during this stage, we have $\max_r \gamma_{j,r,j}^{(t)} \geq m^{1/q}$ for all $j \in [K]$, $\max_{j,r,i}\{|\zeta_{j,r,i}^{(t)}|, |\omega_{j,r,i}^{(t)}|\} \leq 2\beta_1$, $\max_{j\neq k, r \in [m]}\{|\gamma_{j,r,k}^{(t)}|\} \leq 2\beta_2$, and*

$$
\frac{1}{t - T_1} \sum_{s=T_1}^{t} L_S(\widetilde{\mathbf{W}}^{(s)}) \leq \frac{\left\| \widetilde{\mathbf{W}}^{(T_1)} - \widetilde{\mathbf{W}}^{\star} \right\|_F^2}{C\eta(t - T_1)} + \frac{\epsilon}{C}.
$$

*Proof.* From Lemma D.17, we have $max_r \gamma_{j,r,j}^{(T_1)} \geq m^{1/q}$ and since $\gamma^{(t)}$ is an increasing sequence over $t$, we have $max_r \gamma_{j,r,j}^{(t)} \geq m^{1/q}$ for all $t \in [T_1, T_2]$. We have

$$
\left\| \widetilde{\mathbf{W}}^{(s)} - \widetilde{\mathbf{W}}^{\star} \right\|_F^2 - \left\| \widetilde{\mathbf{W}}^{(s+1)} - \widetilde{\mathbf{W}}^{\star} \right\|_F^2 \geq C\eta L_S(\widetilde{\mathbf{W}}^{(s)}) - \eta\epsilon.
$$

Taking a telescopic sum from $T_1$ to $t$ yields

$$\sum_{s=T_1}^{t} L_S(\widetilde{\mathbf{W}}^{(s)}) \leq \frac{\left\|\widetilde{\mathbf{W}}^{(T_1)} - \widetilde{\mathbf{W}}^{\star}\right\|_F^2 + \eta\epsilon(t - T_1)}{C\eta}.$$

Combining Lemma D.18, we have

$$\sum_{s=T_1}^{t} L_S(\widetilde{\mathbf{W}}^{(s)}) \leq O(\eta^{-1}\left\|\widetilde{\mathbf{W}}^{(T_1)} - \widetilde{\mathbf{W}}^{\star}\right\|_F^2) = O(\eta^{-1}Km^3\log^2(1/\epsilon)\mu^{-2}). \tag{7}$$

Define $\Psi^{(t)} = \max_{j,r,i}\{\zeta_{j,r,i}^{(t)}, |\omega_{j,r,i}^{(t)}|\}$ and $\Phi^{(t)} = \max_{j\neq k, r\in[m]}|\gamma_{j,r,k}^{(t)}|$ and $\beta_2 = \widetilde{O}(\sigma_0\mu)$. Now we use induction to prove $\Psi^{(t)} \leq 2\beta_1$ and $\Phi^{(t)} \leq 2\beta_2$. Suppose the result holds for time $t \leq t'$. Then

$$\Psi^{(t+1)} \leq \Psi^{(t)} + \max_{j,r,i}\left\{\frac{\eta}{n}|\ell_{j,i}'^{(t)}| \cdot \sigma'\left(\left\langle\widetilde{\mathbf{w}}_{j,r}^{(0)}, \boldsymbol{\xi}_i\right\rangle + \sum_{k=1}^{K}\gamma_{j,r,k}^{(t)}\frac{\langle\boldsymbol{\mu}_k, \boldsymbol{\xi}_i\rangle}{\|\boldsymbol{\mu}_k\|_2^2} + \sum_{i'=1}^{n}\Psi^{(t)}\frac{\left\langle\widetilde{\boldsymbol{\xi}}_{j,r,i'}, \boldsymbol{\xi}_i\right\rangle}{\left\|\widetilde{\boldsymbol{\xi}}_{j,r,i'}\right\|_2^2}\right)\left\|\widetilde{\boldsymbol{\xi}}_{j,r,i}\right\|_2^2\right\}$$

$$\leq \Psi^{(t)} + \frac{\eta}{n}q\max_i|\ell_{y_i,i}'^{(t)}|\left(O(\sqrt{\log d}\sigma_0\sigma_n\sqrt{pd}) + K\log^{1/q}T^{\star}\frac{\mu\sigma_n\sqrt{\log d}}{\mu^2}\right.$$

$$\left. + \frac{O(\sigma_n^2 pd) + nO(\sigma_n^2\sqrt{pd\log d})}{\Theta(\sigma_n^2 pd)}\Psi^{(t)}\right)^{q-1}O(\sigma_n^2 pd)$$

$$\leq \Psi^{(t)} + \frac{\eta}{n}\sum_{i=1}^{n}|\ell_{y_i,i}'^{(t)}|\left(O(\sqrt{\log d}\sigma_0\sigma_n\sqrt{pd}) + O(\Psi^{(t)})\right)^{q-1}O(\sigma_n^2 pd),$$

where the second inequality follows by $|\ell_{j,i}'^{(t)}| \leq |\ell_{y_i,i}'^{(t)}|$ and applying the bounds from Lemma D.5, and the last inequality follows by choosing $\frac{K\log^{1/q}T^{\star}}{\sqrt{d}} = \widetilde{O}(\frac{1}{\sqrt{d}}) \ll \sigma_0\sigma_n\sqrt{pd}$. Unrolling the recursion by taking a sum from $T_1$ to $t'$ we have

$$\Psi^{(t'+1)} \overset{(i)}{\leq} \Psi^{(T_1)} + \frac{\eta}{n}\sum_{s=T_1}^{t'}\sum_{i=1}^{n}|\ell_{y_i,i}'^{(s)}|O(\sigma_n^2 pd\,\text{poly}\log d)\beta_1^{q-1}$$

$$\overset{(ii)}{\leq} \Psi^{(T_1)} + \frac{\eta}{n}O(\sigma_n^2 pd\,\text{poly}\log d)\beta_1^{q-1}\sum_{s=T_1}^{t'}\sum_{i=1}^{n}\ell_i^{(s)}$$

$$= \Psi^{(T_1)} + \frac{\eta}{n}O(\sigma_n^2 pd\,\text{poly}\log d)\beta_1^{q-1}\sum_{s=T_1}^{t'}L_S(\widetilde{\mathbf{W}}^{(s)})$$

$$\overset{(iii)}{\leq} \Psi^{(T_1)} + \frac{1}{n}O(Km^3\mu^{-2}\sigma_n^2 pd\,\text{poly}\log d)\beta_1^{q-1}$$

$$\overset{(iv)}{\leq} \beta_1 + \widetilde{O}(Km^3)\beta_1^{q-1}$$

$$\overset{(v)}{\leq} 2\beta_1,$$

where (i) follows from induction hypothesis $\Psi^{(t)} \leq 2\beta_1$, (ii) follows from the property of cross-entropy loss with softmax $|\ell_{j,i}'| \leq |\ell_{y_i,i}'| \leq \ell_i$, (iii) follows from Equation equation 7, (iv) follows from our choice of $\mu, n, K$, and (v) follows because $\widetilde{O}(Km^3)\beta_1^{q-2} \leq \widetilde{O}(Km^3\sigma_0\sigma_n\sqrt{pd}) \leq 1$. Therefore, by induction $\Psi^{(t)} \leq 2\beta_1$ holds for time $t \leq t' + 1$.

On the other hand,

$$\Phi^{(t'+1)}$$

$$\overset{(i)}{\leq} \Phi^{(t)} + \max_{j,r,k,i} \left\{ \frac{\eta}{n} \sum_{i=1}^{n} \mathbb{I}(y_i = k) |\ell_{j,i}'^{(t)}| \sigma' \left( \left\langle \widetilde{\mathbf{w}}_{j,r}^{(0)}, \boldsymbol{\mu}_k \right\rangle + \sum_{i'=1}^{n} \zeta_{j,r,i'}^{(t)} \frac{\left\langle \widetilde{\boldsymbol{\xi}}_{j,r,i'}, \boldsymbol{\mu}_k \right\rangle}{\left\| \widetilde{\boldsymbol{\xi}}_{j,r,i'} \right\|_2^2} + \sum_{i'=1}^{n} \omega_{j,r,i'}^{(t)} \frac{\left\langle \widetilde{\boldsymbol{\xi}}_{j,r,i'}, \boldsymbol{\mu}_k \right\rangle}{\left\| \widetilde{\boldsymbol{\xi}}_{j,r,i'} \right\|_2^2} \right) \mu^2 \right\}$$

$$\overset{(ii)}{\leq} \Phi^{(t)} + \Theta(\frac{\eta}{K}) \max_{j,i} |\ell_{j,i}'^{(t)}| \left( O(\sigma_0 \mu \sqrt{\log d}) + n O(\sigma_0 \sigma_n \sqrt{pd}) \frac{\sigma_n \mu \sqrt{\log d}}{\sigma_n^2 pd} \right)^{q-1} \mu^2$$

$$\overset{(iii)}{\leq} \Phi^{(T_1)} + \Theta(\frac{\eta}{K}) \mu^2 \sum_{s=T_1}^{t} \sum_{i=1}^{n} \ell_i^{(s)} \left( O(\sigma_0 \mu \sqrt{\log d}) \right)^{q-1}$$

$$\overset{(iv)}{\leq} \beta_2 + O(m^3) \beta_2^{q-1}$$

$$\overset{(v)}{\leq} 2\beta_2,$$

where (i) follows because $\gamma_{j,r,k}^{(t)} \leq 0$, (ii) follows from Lemma D.7 and Lemma D.5, (iii) follows because $\max_{j,i} |\ell_{j,i}'^{(t)}| \leq \max_i |\ell_{y_i,i}'^{(t)}| \leq \max_i \ell_i^{(t)} \leq \sum_i \ell_i^{(t)}$, (iv) follows from Equation equation 7, and (v) follows because $O(m^3) \beta_2^{q-2} \leq \widetilde{O}(m^3 \sigma_0 \mu) \leq 1$. □

### D.5 GENERALIZATION ANALYSIS

In this subsection, we show that pruning can purify the feature by reducing the variance of the noise by a factor of $p$ when a new sample is given.

Now the network has parameter

$$\widetilde{\mathbf{w}}_{j,r}^{\star} = \widetilde{\mathbf{w}}_{j,r}^{(0)} + \sum_{k=1}^{K} \gamma_{j,r,k}^{\star} \frac{\boldsymbol{\mu}_k \odot \mathbf{m}_{j,r}}{\mu^2} + \sum_{i=1}^{n} \zeta_{j,r,i}^{\star} \frac{\widetilde{\boldsymbol{\xi}}_{j,r,i}}{\left\| \widetilde{\boldsymbol{\xi}}_{j,r,i} \right\|_2^2} + \sum_{i=1}^{n} \omega_{j,r,i}^{\star} \frac{\widetilde{\boldsymbol{\xi}}_{j,r,i}}{\left\| \widetilde{\boldsymbol{\xi}}_{j,r,i} \right\|_2^2}.$$

We have $\left\| \widetilde{\mathbf{w}}_{j,r}^{\star} \right\|_2 = O(\sigma_0 \sqrt{pd} + \mu^{-1} \log^{1/q}(T^{\star}) + K\sigma_0 \operatorname{poly} \log d + n\sigma_0 \sigma_n \sqrt{pd} \frac{1}{\sigma_n \sqrt{pd}}) = O(\sigma_0 \sqrt{pd})$.

**Lemma D.25** (Formal Restatement of Lemma 3.6). *With probability at least* $1 - 2Km \exp\left( -\frac{(2m)^{-4/q}}{O(\sigma_0^2 \sigma_n^2 pd)} \right)$,

$$\max_{j,r} \left| \left\langle \widetilde{\mathbf{w}}_{j,r}^{\star}, \boldsymbol{\xi} \right\rangle \right| \leq (2m)^{-2/q}.$$

*Proof.* Since $\left\langle \widetilde{\mathbf{w}}_{j,r}^{\star}, \boldsymbol{\xi} \right\rangle$ follows a Gaussian distribution with variance $O(\sigma_0^2 \sigma_n^2 pd)$, we have

$$\mathbb{P}\left[ \left| \left\langle \widetilde{\mathbf{w}}_{j,r}^{\star}, \boldsymbol{\xi} \right\rangle \right| \geq (2m)^{-2/q} \right] \leq 2 \exp\left( -\frac{(2m)^{-4/q}}{O(\sigma_0^2 \sigma_n^2 pd)} \right).$$

Applying a union bound over $j \in [K], r \in [m]$ gives the result. □

**Theorem D.26** (Formal Restatement of Generalization Part of Theorem 3.1). *Under the same assumptions as Theorem D.1, within* $\widetilde{O}(K\eta^{-1}\sigma_0^{2-q}\mu^{-q} + K^2 m^4 \mu^{-2} \eta^{-1} \epsilon^{-1})$ *iterations, we can find* $\widetilde{\mathbf{W}}^{\star}$ *such that*

- $L_S(\widetilde{\mathbf{W}}^{\star}) \leq \epsilon$.

- $L_{\mathcal{D}} \leq O(K\epsilon) + \exp(-n^2/p)$.

*Proof.* Let $\mathcal{E}$ be the event that Lemma D.25 holds. Then, we can divide $L_{\mathcal{D}}(\widetilde{\mathbf{W}}^{\star})$ into two parts:

$$\mathbb{E}[\ell(F(\widetilde{\mathbf{W}}^{\star}, \mathbf{x}))] = \underbrace{\mathbb{E}[\mathbb{I}(\mathcal{E})\ell(F(\widetilde{\mathbf{W}}^{\star}, \mathbf{x}))]}_{I_1} + \underbrace{\mathbb{E}[\mathbb{I}(\mathcal{E}^c)\ell(F(\widetilde{\mathbf{W}}^{\star}, \mathbf{x}))]}_{I_2}.$$

Since $L_S(\widetilde{\mathbf{W}}^\star) \leq \epsilon$, for each class $j \in [K]$ there must exist one training sample $(\mathbf{x}_i, y_i) \in S$ with $y_i = j$ such that $\ell(F(\widetilde{\mathbf{W}}^\star, \mathbf{x}_i)) \leq K\epsilon \leq 1$ by pigeonhole principle. This implies that $\sum_{j' \neq j} \exp(F_{j'}(\mathbf{x}_i) - F_j(\mathbf{x}_i)) \leq 2K\epsilon$. Conditioning on the event $\mathcal{E}$, by Lemma D.25, we have

$$|F_j(\widetilde{\mathbf{W}}^\star, \mathbf{x}) - F_j(\widetilde{\mathbf{W}}^\star, \mathbf{x}_i)| \leq \sum_r \sigma(\langle \widetilde{\mathbf{w}}_{j,r}^\star, \boldsymbol{\xi}_i \rangle) + \sum_r \sigma(\langle \widetilde{\mathbf{w}}_{j,r}^\star, \boldsymbol{\xi} \rangle)$$
$$\leq \sum_r (2m)^{-1} + \sum_r (2m)^{-1}$$
$$\leq 1.$$

Thus, we have $\exp(F_{j'}(\mathbf{x}) - F_j(\mathbf{x})) \leq 2K\epsilon e^2 = O(K\epsilon)$. Next we bound the term $I_2$.

$$\ell(F(\widetilde{\mathbf{W}}^\star, \mathbf{x})) = \log\left(1 + \sum_{j' \neq y} \exp(F_{j'}(\mathbf{x}) - F_y(\mathbf{x}))\right)$$
$$\leq \log\left(1 + \sum_{j' \neq y} \exp(F_{j'}(\mathbf{x}))\right)$$
$$\leq \sum_{j' \neq y} \log(1 + \exp(F_{j'}(\mathbf{x})))$$
$$\leq K + \sum_{j' \neq y} F_{j'}(\mathbf{x})$$
$$= K + \sum_{j' \neq y} \sigma(\langle \widetilde{\mathbf{w}}_{j',r}^\star, \boldsymbol{\mu}_y \rangle) + \sigma(\langle \widetilde{\mathbf{w}}_{j',r}^\star, \boldsymbol{\xi} \rangle)$$
$$\leq K + Km(O(\sigma_0 \mu \sqrt{\log d}))^q + \widetilde{O}(m(\sigma_0 \sigma_n \sqrt{d})^q) \|\boldsymbol{\xi}/\sigma_n\|_2^q$$
$$\leq 2K + \|\boldsymbol{\xi}/\sigma_n\|_2^q, \tag{8}$$

where the first inequality follows because $F_y(\mathbf{x}) \geq 0$, the second and third inequalities follow from the property of log function, and the last inequality follows from our choice of $\sigma_0 \leq \widetilde{O}(m^{-4}n^{-1}\sigma_n^{-1}d^{-1/2})$. We further have

$$I_2 \leq \sqrt{\mathbb{E}[\mathbb{I}(\mathcal{E})]}\sqrt{\mathbb{E}[\ell(F(\widetilde{\mathbf{W}}^\star, \mathbf{x}))^2]}$$
$$\leq \sqrt{\mathbb{P}(\mathcal{E}^c)}\sqrt{4K^2 + \mathbb{E}\|\boldsymbol{\xi}/\sigma_n\|_2^{2q}}$$
$$\leq \exp(-Cm^{-2/q}\sigma_0^{-2}\sigma_n^{-2}p^{-1}d^{-1} + \log(d))$$
$$\leq \exp(-n^2/p),$$

where the first inequality follows from Cauchy-Schwarz inequality, the second inequality follows from Equation equation 8, the third inequality follows from Lemma D.25, and the last inequality follows because $\sigma_0 \leq \widetilde{O}(m^{-4}n^{-1}\sigma_n^{-1}d^{-1/2})$.

$\square$

## E  PROOF OF THEOREM 4.1

In this section, we show that there exists a relatively large pruning fraction (i.e., small $p$) such that while gradient descent is still able to drive the training error toward zero, the learned model yields poor generalization. We first provide a formal restatement of Theorem 4.1.

**Theorem E.1** (Formal Restatement of Theorem 4.1). *Under Condition 2.2, choose initialization variance $\sigma_0 = \widetilde{\Theta}(m^{-4}n^{-1}\mu^{-1})$ and learning rate $\eta \leq \widetilde{O}(1/\mu^2)$. For $\epsilon > 0$, if $p = \Theta(\frac{1}{Km\log d})$, then with probability at least $1 - 1/\log(d)$, there exists $T = O(\eta^{-1}n\sigma_0^{q-2}\sigma_n^{-q}(pd)^{-q/2} + \eta^{-1}\epsilon^{-1}m^4n\sigma_n^{-2}(pd)^{-1})$ such that the following holds:*

   *1. The training loss is below $\epsilon$: $L_S(\widetilde{\mathbf{W}}^{(T)}) \leq \epsilon$.*

2. *The model weight doesn't learn any of its corresponding signal at all:* $\gamma_{j,r,j}^{(t)} = 0$ *for all* $j \in [K]$, $r \in [m]$.

3. *The model weights is highly correlated with the noise:* $\max_{r \in [m]} \zeta_{j,r,i}^{(T)} \geq \Omega(m^{-1/q})$ *if* $y_i = j$.

*Moreover, the testing loss is large:*

$$L_{\mathcal{D}}(\widetilde{\mathbf{W}}^{(T)}) \geq \Omega(\log K).$$

The proof of Theorem 4.1 consists of the analysis of the over-pruning for three stages of gradient descent: initialization, feature growing phase, and converging phase, and the establishment of the generalization property. We present these analysis in detail in the following subsections.

### E.1 INITIALIZATION

**Lemma E.2.** *When* $m = \text{poly} \log d$ *and* $p = \Theta(\frac{1}{Km \log d})$, *with probability* $1 - O(1/\log d)$, *for all class* $j \in [K]$ *we have* $|\mathcal{S}_{\text{signal}}^j| = 0$.

*Proof.* First, the probability that a given class $j$ receives no signal is $(1-p)^m$. We use the inequality that

$$1 + t \geq \exp\{O(t)\} \quad \forall t \in (-1/4, 1/4).$$

Then the probability that $|\mathcal{S}_{\text{signal}}^j| = 0$, $\forall j \in [K]$ is given by

$$(1-p)^{Km} \geq \exp\{-O(pKm)\} \geq 1 - O\left(\frac{1}{\log d}\right).$$

$\square$

### E.2 FEATURE GROWING PHASE

**Lemma E.3** (Formal Restatement of Lemma 4.3). *Under the same assumption as Theorem E.1, there exists* $T_1 < T^\star$ *such that* $T_1 = O(\eta^{-1} n \sigma_0^{q-2} \sigma_n^{-q}(pd)^{-q/2})$ *and we have*

- $\max_r \zeta_{y_i,r,i} \geq m^{-1/q}$ *for all* $i \in [n]$.

- $\max_{j,r,i} |\omega_{j,r,i}^{(t)}| = \widetilde{O}(\sigma_0 \sigma_n \sqrt{pd})$.

- $\max_{j,r,k} |\gamma_{j,r,k}^{(t)}| \leq \widetilde{O}(\sigma_0 \mu)$.

*Proof.* First of all, recall that from Definition C.1 we have for $j = y_i$

$$\left\langle \widetilde{\mathbf{w}}_{j,r}^{(t)}, \boldsymbol{\xi}_i \right\rangle$$

$$= \left\langle \widetilde{\mathbf{w}}_{j,r}^{(0)}, \boldsymbol{\xi}_i \right\rangle + \zeta_{j,r,i}^{(t)} + \sum_{k \neq j} \gamma_{j,r,k}^{(t)} \frac{\left\langle \boldsymbol{\mu}_k, \widetilde{\boldsymbol{\xi}}_{j,r,i} \right\rangle}{\mu^2} + \sum_{i' \neq i} \zeta_{j,r,i}^{(t)} \frac{\left\langle \widetilde{\boldsymbol{\xi}}_{j,r,i'}, \boldsymbol{\xi}_i \right\rangle}{\left\| \widetilde{\boldsymbol{\xi}}_{j,r,i'} \right\|_2^2} + \sum_{i'=1}^n \omega_{j,r,i}^{(t)} \frac{\left\langle \widetilde{\boldsymbol{\xi}}_{j,r,i'}, \boldsymbol{\xi}_i \right\rangle}{\left\| \widetilde{\boldsymbol{\xi}}_{j,r,i'} \right\|_2^2}.$$

Let

$$B_i^{(t)} = \max_{j=y_i,r} \left\{ \zeta_{j,r,i}^{(t)} + \left\langle \widetilde{\mathbf{w}}_{j,r}^{(0)}, \boldsymbol{\xi}_i \right\rangle - O(n \log^{1/q} T^\star \sqrt{\frac{\log d}{pd}}) - O(n \sigma_0 \sigma_n \sqrt{pd} \sqrt{\frac{\log d}{pd}}) \right\}.$$

Since $\max_{j=y_i,r} \left\langle \widetilde{\mathbf{w}}_{j,r}^{(0)}, \boldsymbol{\xi}_i \right\rangle \geq \Omega(\sigma_0 \sigma_n \sqrt{pd})$, we have

$$B_i^{(0)} \geq \Omega(\sigma_0 \sigma_n \sqrt{pd}) - O(n \log^{1/q} T^\star \sqrt{\frac{\log d}{pd}}) - O(n \sigma_0 \sigma_n \sqrt{pd} \sqrt{\frac{\log d}{pd}}) \geq \Omega(\sigma_0 \sigma_n \sqrt{pd}).$$

Let $T_i$ to be the last time that $\zeta_{j,r,i}^{(t)} \leq m^{-1/q}$. We can compute the growth of $B_i^{(t)}$ as

$$
\begin{aligned}
B_i^{(t+1)} &\geq B_i^{(t)} + \Theta(\frac{\eta \sigma_n^2 pd}{n})[B_i^{(t)}]^{q-1} \\
&\geq B_i^{(t)} + \Theta(\frac{\eta \sigma_n^2 pd}{n})[B_i^{(0)}]^{q-2} B_i^{(t)} \\
&\geq \left(1 + \Theta\left(\frac{\eta \sigma_0^{q-2} \sigma_n^q p^{q/2} d^{q/2}}{n}\right)\right) B_i^{(t)}.
\end{aligned}
$$

Therefore, $B_i^{(t)}$ will reach $2m^{-1/q}$ within $\widetilde{O}(\eta^{-1} n \sigma_0^{q-2} \sigma_n^{-q}(pd)^{-q/2})$ iterations.

On the other hand, by Proposition D.10, we have $|\omega_{j,r,i}^{(t)}| \leq \beta + 6Cn\alpha\sqrt{\frac{\log d}{pd}} = O(\sigma_0 \sigma_n \sqrt{pd \log d})$. $\qquad\square$

### E.3 Converging Phase

From the first stage we know that

$$
\widetilde{\mathbf{w}}_{j,r}^{(T_1)} = \widetilde{\mathbf{w}}_{j,r}^{(0)} + + \sum_{k \neq j} \gamma_{j,r,k}^{(t)} \frac{\boldsymbol{\mu}_k \odot \mathbf{m}_{j,r}}{\mu^2} + \sum_{i=1}^n \zeta_{j,r,i}^{(T_1)} \frac{\widetilde{\boldsymbol{\xi}}_{j,r,i}}{\left\|\widetilde{\boldsymbol{\xi}}_{j,r,i}\right\|_2^2} + \sum_{i=1}^n \omega_{j,r,i}^{(T_1)} \frac{\widetilde{\boldsymbol{\xi}}_{j,r,i}}{\left\|\widetilde{\boldsymbol{\xi}}_{j,r,i}\right\|_2^2}.
$$

Now we define $\widetilde{\mathbf{W}}^\star$ as follows:

$$
\widetilde{\mathbf{w}}_{j,r}^\star = \widetilde{\mathbf{w}}_{j,r}^{(0)} + \Theta(m \log(1/\epsilon)) \left[\sum_{i=1}^n \mathbb{I}(j = y_i) \frac{\widetilde{\boldsymbol{\xi}}_{j,r,i}}{\left\|\widetilde{\boldsymbol{\xi}}_{j,r,i}\right\|_2^2}\right].
$$

**Lemma E.4.** *Based on the result from feature growing phase, $\left\|\widetilde{\mathbf{W}}^{(T_1)} - \widetilde{\mathbf{W}}^\star\right\|_F \leq O(m^2 n^{1/2} \log(1/\epsilon) \sigma_n^{-1}(pd)^{-1/2})$.*

*Proof.* We derive the following bound:

$$
\begin{aligned}
&\left\|\widetilde{\mathbf{W}}^{(T_1)} - \widetilde{\mathbf{W}}^\star\right\|_F \\
&\leq \left\|\widetilde{\mathbf{W}}^{(T_1)} - \widetilde{\mathbf{W}}^{(0)}\right\|_F + \left\|\widetilde{\mathbf{W}}^{(0)} - \widetilde{\mathbf{W}}^\star\right\|_F \\
&\leq \sum_{j,r} \left(\left\|\sum_{k \neq j} \gamma_{j,r,k}^{(t)} \frac{\boldsymbol{\mu}_k}{\mu^2}\right\|_2 + \left\|\sum_{i=1}^n \zeta_{j,r,i}^{(T_1)} \frac{\widetilde{\boldsymbol{\xi}}_{j,r,i}}{\left\|\widetilde{\boldsymbol{\xi}}_{j,r,i}\right\|_2^2}\right\|_2 + \left\|\sum_{i=1}^n \omega_{j,r,i}^{(T_1)} \frac{\widetilde{\boldsymbol{\xi}}_{j,r,i}}{\left\|\widetilde{\boldsymbol{\xi}}_{j,r,i}\right\|_2^2}\right\|_2\right) + \Theta(m^2 n^{1/2} \log(1/\epsilon) \sigma_n^{-1}(pd)^{-1/2}) \\
&\leq Km(O(\sqrt{K}\sigma_0) + O(n^{1/2}\sigma_n^{-1}(pd)^{-1/2}\log^{1/q} T^\star)) + \widetilde{O}(m^2 n^{1/2} \log(1/\epsilon)\sigma_n^{-1}(pd)^{-1/2}) \\
&\leq \widetilde{O}(m^2 n^{1/2} \log(1/\epsilon)\sigma_n^{-1}(pd)^{-1/2}),
\end{aligned}
$$

where the first inequality follows from triangle inequality, the second inequality follows from the expression of $\mathbf{W}^{(T_1)}, \mathbf{W}^\star$, and the third inequality follows from Lemma D.5 and the fact that $\zeta_{j,r,i}^{(t)} > 0$ if and only if $j = y_i$. $\qquad\square$

**Lemma E.5.** *For $T_1 \leq t \leq T^\star$, we have*

$$
\left\langle \nabla F_{y_i}(\widetilde{\mathbf{W}}_{y_i}, \mathbf{x}_i), \widetilde{\mathbf{W}}_{y_i}^\star \right\rangle - \left\langle \nabla F_j(\widetilde{\mathbf{W}}_j, \mathbf{x}_i), \widetilde{\mathbf{W}}_j^\star \right\rangle \geq q \log \frac{2qK}{\epsilon}.
$$

**Lemma E.6.** *For $T_1 \leq t \leq T^\star$ and $j = y_i$, we have*

$$
\left\langle \nabla F_j(\widetilde{\mathbf{W}}_j^{(t)}, \mathbf{x}_i), \widetilde{\mathbf{W}}_j^\star \right\rangle \geq \Theta(m^{1/q} \log(1/\epsilon)).
$$

*Proof.* By Lemma D.5, we have $\left\langle \widetilde{\boldsymbol{\xi}}_{j,r,i}, \widetilde{\mathbf{w}}_{j,r}^{\star} \right\rangle = \Theta(m \log(1/\epsilon))$ and by Lemma E.3 for $j = y_i$, $\max_r \left\langle \widetilde{\mathbf{w}}_{j,r}^{(t)}, \boldsymbol{\xi}_i \right\rangle \geq \max_r \zeta_{j,r,i} - \max_r \left\langle \widetilde{\mathbf{w}}_{j,r}^{(0)}, \boldsymbol{\xi}_i \right\rangle - O(n\sqrt{\frac{\log d}{d}}\alpha) \geq \Theta(m^{-1/q})$. Then we have

$$
\begin{aligned}
\left\langle \nabla F_j(\widetilde{\mathbf{W}}_j^{(t)}, \mathbf{x}_i), \widetilde{\mathbf{W}}_j^{\star} \right\rangle &= \sum_{r=1}^{m} \sigma' \left( \left\langle \widetilde{\mathbf{w}}_{j,r}^{(t)}, \boldsymbol{\xi}_i \right\rangle \right) \left\langle \widetilde{\boldsymbol{\xi}}_{j,r,i}, \widetilde{\mathbf{w}}_{j,r}^{\star} \right\rangle \\
&\geq \Theta(m^{1/q} \log(1/\epsilon)).
\end{aligned}
$$

$\square$

**Lemma E.7.** *For $T_1 \leq t \leq T^{\star}$ and $j \neq y_i$, we have*

$$
\left\langle \nabla F_j(\widetilde{\mathbf{W}}_j^{(t)}, \mathbf{x}_i), \widetilde{\mathbf{W}}_j^{\star} \right\rangle \leq O(1).
$$

*Proof.* We first compute $\left\langle \widetilde{\mathbf{w}}_{j,r}^{\star}, \boldsymbol{\xi}_i \right\rangle = \left\langle \widetilde{\mathbf{w}}_{j,r}^{(0)}, \boldsymbol{\xi}_i \right\rangle + \Theta(m \log(1/\epsilon)) \sum_{i=1}^{n} \mathbb{I}(j = y_i) \frac{\langle \widetilde{\boldsymbol{\xi}}_{j,r,i}, \boldsymbol{\xi}_i \rangle}{\left\| \widetilde{\boldsymbol{\xi}}_{j,r,i} \right\|_2^2} = O(\sigma_0 \sigma_n \sqrt{pd \log d})$. Further,

$$
\begin{aligned}
&\left\langle \widetilde{\mathbf{w}}_{j,r}^{(t)}, \boldsymbol{\xi}_i \right\rangle \\
&= \left\langle \widetilde{\mathbf{w}}_{j,r}^{(0)}, \boldsymbol{\xi}_i \right\rangle + \sum_{k \neq j} \gamma_{j,r,k}^{(t)} \frac{\left\langle \boldsymbol{\mu}_k, \widetilde{\boldsymbol{\xi}}_{j,r,i} \right\rangle}{\mu^2} + \sum_{i=1}^{n} \zeta_{j,r,i}^{(t)} \frac{\left\langle \widetilde{\boldsymbol{\xi}}_{j,r,i}, \boldsymbol{\xi}_i \right\rangle}{\left\| \widetilde{\boldsymbol{\xi}}_{j,r,i} \right\|_2^2} + \sum_{i=1}^{n} \omega_{j,r,i}^{(t)} \frac{\left\langle \widetilde{\boldsymbol{\xi}}_{j,r,i}, \boldsymbol{\xi}_i \right\rangle}{\left\| \widetilde{\boldsymbol{\xi}}_{j,r,i} \right\|_2^2} \\
&\leq O(\sigma_0 \sigma_n \sqrt{pd \log d}),
\end{aligned}
$$

where the inequality follows from Lemma D.5 and Lemma D.15. Thus, we have

$$
\begin{aligned}
\left\langle \nabla F_j(\widetilde{\mathbf{W}}_j^{(t)}, \mathbf{x}_i), \widetilde{\mathbf{W}}_j^{\star} \right\rangle &= \sum_{r=1}^{m} \sigma' \left( \left\langle \widetilde{\mathbf{w}}_{j,r}^{(t)}, \boldsymbol{\xi}_i \right\rangle \right) \left\langle \widetilde{\boldsymbol{\xi}}_{j,r,i}, \widetilde{\mathbf{w}}_{j,r}^{\star} \right\rangle \\
&\leq m O \left( \sigma_0 \sigma_n \sqrt{pd \log d} \right)^q \\
&\leq O(1),
\end{aligned}
$$

where the last inequality follows from our choice of $\sigma_0 \leq \widetilde{O}(m^{-1/q}\mu^{-1})$. $\square$

**Lemma E.8.** *Under the same assumption as Theorem E.1, we have*

$$
\left\| \mathbf{W}^{(t)} - \mathbf{W}^{\star} \right\|_F^2 - \left\| \mathbf{W}^{(t+1)} - \mathbf{W}^{\star} \right\|_F^2 \geq C\eta L_S(\widetilde{\mathbf{W}}^{(t)}) - \eta\epsilon.
$$

*Proof.* To simplify our notation, we define $\widehat{F}_j^{(t)}(\mathbf{x}_i) = \left\langle \nabla F_j(\widetilde{\mathbf{W}}_j^{(t)}, \mathbf{x}_i), \widetilde{\mathbf{W}}_j^\star \right\rangle$. The proof is exactly the same as the proof of Lemma D.23.

$$
\left\| \widetilde{\mathbf{W}}^{(t)} - \widetilde{\mathbf{W}}^\star \right\|_F^2 - \left\| \widetilde{\mathbf{W}}^{(t+1)} - \widetilde{\mathbf{W}}^\star \right\|_F^2
$$

$$
= 2\eta \left\langle \nabla L_S(\widetilde{\mathbf{W}}^{(t)}) \odot \mathbf{M}, \widetilde{\mathbf{W}}^{(t)} - \widetilde{\mathbf{W}}^\star \right\rangle - \eta^2 \left\| \nabla L_S(\widetilde{\mathbf{W}}^{(t)}) \odot \mathbf{M} \right\|_F^2
$$

$$
= \frac{2\eta}{n} \sum_{i=1}^n \sum_{j=1}^K \ell_{j,i}'^{(t)} \left[ q F_j(\widetilde{\mathbf{W}}_j^{(t)}; \mathbf{x}_i, y_i) - \left\langle \nabla F_j(\widetilde{\mathbf{W}}_j^{(t)}, \mathbf{x}_i), \widetilde{\mathbf{W}}_j^\star \right\rangle \right] - \eta^2 \left\| \nabla L_S(\widetilde{\mathbf{W}}^{(t)}) \odot \mathbf{M} \right\|_F^2
$$

$$
\geq \frac{2q\eta}{n} \sum_{i=1}^n \left[ \log(1 + \sum_{j=1}^K e^{F_j - F_{y_i}}) - \log(1 + \sum_{j=1}^K e^{(\widehat{F}_j - \widehat{F}_{y_i})/q}) \right] - \eta^2 \left\| \nabla L_S(\widetilde{\mathbf{W}}^{(t)}) \odot \mathbf{M} \right\|_F^2
$$

$$
\geq \frac{2q\eta}{n} \sum_{i=1}^n \left[ \ell(\widetilde{\mathbf{W}}^{(t)}; \mathbf{x}_i, y_i) - \log(1 + K e^{-\log(2qK/\epsilon)}) \right] - \eta^2 \left\| \nabla L_S(\widetilde{\mathbf{W}}^{(t)}) \odot \mathbf{M} \right\|_F^2
$$

$$
\geq \frac{2q\eta}{n} \sum_{i=1}^n \left[ \ell(\widetilde{\mathbf{W}}^{(t)}; \mathbf{x}_i, y_i) - \frac{\epsilon}{2q} \right] - \eta^2 \left\| \nabla L_S(\widetilde{\mathbf{W}}^{(t)}) \odot \mathbf{M} \right\|_F^2
$$

$$
\geq C\eta L_S(\widetilde{\mathbf{W}}^{(t)}) - \eta\epsilon,
$$

where the first inequality follows from the convexity of the cross-entropy loss with softmax, the second inequality follows from Lemma D.20, the third inequality follows because $\log(1 + x) \leq x$, and the last inequality follows from Lemma D.19 for some constant $C > 0$. □

**Lemma E.9** (Formal Restatement of Lemma 4.4). *Under the same assumption as Theorem E.1, choose* $T_2 = T_1 + \lceil \frac{\left\| \widetilde{\mathbf{W}}^{(T_1)} - \widetilde{\mathbf{W}}^\star \right\|_F^2}{2\eta\epsilon} \rceil = T_1 + \widetilde{O}(\eta^{-1}\epsilon^{-1}m^4 n\sigma_n^{-2}(pd)^{-1})$. *Then for any time* $t$ *during this stage we have* $\max_{j,r} |\omega_{j,r,i}^{(t)}| = O(\sigma_0 \sqrt{pd})$ *and*

$$
\frac{1}{t - T_1} \sum_{s=T_1}^t L_S(\widetilde{\mathbf{W}}^{(s)}) \leq \frac{\left\| \widetilde{\mathbf{W}}^{(T_1)} - \widetilde{\mathbf{W}}^\star \right\|_F^2}{C\eta(t - T_1)} + \frac{\epsilon}{C}.
$$

*Proof.* We have

$$
\left\| \widetilde{\mathbf{W}}^{(s)} - \widetilde{\mathbf{W}}^\star \right\|_F^2 - \left\| \widetilde{\mathbf{W}}^{(s+1)} - \widetilde{\mathbf{W}}^\star \right\|_F^2 \geq C\eta L_S(\widetilde{\mathbf{W}}^{(s)}) - \eta\epsilon.
$$

Taking a telescopic sum from $T_1$ to $t$ yields

$$
\sum_{s=T_1}^t L_S(\widetilde{\mathbf{W}}^{(s)}) \leq \frac{\left\| \widetilde{\mathbf{W}}^{(T_1)} - \widetilde{\mathbf{W}}^\star \right\|_F^2 + \eta\epsilon(t - T_1)}{C\eta}.
$$

Combining Lemma E.4, we have

$$
\sum_{s=T_1}^t L_S(\widetilde{\mathbf{W}}^{(s)}) \leq O(\eta^{-1} \left\| \widetilde{\mathbf{W}}^{(T_1)} - \widetilde{\mathbf{W}}^\star \right\|_F^2) = \widetilde{O}(\eta^{-1}m^4 n\sigma_n^{-2}(pd)^{-1}).
$$

□

## E.4 GENERALIZATION ANALYSIS

**Theorem E.10** (Formal Restatement of the Generalization Part of Theorem 4.1). *Under the same assumption as Theorem E.1, within* $O(\eta^{-1}n\sigma_0^{q-2}\sigma_n^{-q}(pd)^{-q/2} + \eta^{-1}\epsilon^{-1}m^4 n\sigma_n^{-2}(pd)^{-1})$ *iterations, we can find* $\widetilde{\mathbf{W}}^{(T)}$ *such that* $L_S(\widetilde{\mathbf{W}}^{(T)}) \leq \epsilon$, *and* $L_{\mathcal{D}}(\widetilde{\mathbf{W}}^{(t)}) \geq \Omega(\log K)$.

*Proof.* First of all, from Lemma E.9 we know there exists $t \in [T_1, T_2]$ such that $L_S(\widetilde{\mathbf{W}}^{(T)}) \leq \epsilon$. Then, we can bound

$$\left\| \widetilde{\mathbf{w}}_{j,r}^{(t)} \right\|_2 = \left\| \widetilde{\mathbf{w}}_{j,r}^{(0)} + \sum_{k \neq j} \gamma_{j,r,k}^{(t)} \frac{\boldsymbol{\mu}_k}{\mu^2} + \sum_{i=1}^{n} \zeta_{j,r,i}^{(t)} \frac{\widetilde{\boldsymbol{\xi}}_{j,r,i}}{\left\| \widetilde{\boldsymbol{\xi}}_{j,r,i} \right\|_2^2} + \sum_{i=1}^{n} \omega_{j,r,i}^{(t)} \frac{\widetilde{\boldsymbol{\xi}}_{j,r,i}}{\left\| \widetilde{\boldsymbol{\xi}}_{j,r,i} \right\|_2^2} \right\|_2$$

$$\leq \left\| \widetilde{\mathbf{w}}_{j,r}^{(0)} \right\|_2 + \sum_{k \neq j} |\gamma_{j,r,k}^{(t)}| \frac{1}{\mu} + \sum_{i=1}^{n} \zeta_{j,r,i}^{(t)} \frac{1}{\left\| \widetilde{\boldsymbol{\xi}}_{j,r,i} \right\|_2} + \sum_{i=1}^{n} |\omega_{j,r,i}^{(t)}| \frac{1}{\left\| \widetilde{\boldsymbol{\xi}}_{j,r,i} \right\|_2}$$

$$\leq O(\sigma_0 \sqrt{d}) + \widetilde{O}(n \sigma_n^{-1} (pd)^{-1/2}).$$

Consider a new example $(\mathbf{x}, y)$. Taking a union bound over $r$, with probability at least $1 - d^{-1}$, we have

$$\left| \left\langle \mathbf{w}_{y,r}^{(t)}, \boldsymbol{\xi} \right\rangle \right| = \widetilde{O}(\sigma_0 \sigma_n \sqrt{d} + n(pd)^{-1/2}),$$

for all $r \in [m]$. Then,

$$F_y(\mathbf{x}) = \sum_{r=1}^{m} \sigma \left( \left\langle \widetilde{\mathbf{w}}_{j,r}^{(t)}, \boldsymbol{\mu}_y \right\rangle \right) + \sigma \left( \left\langle \widetilde{\mathbf{w}}_{j,r}^{(t)}, \boldsymbol{\xi} \right\rangle \right)$$

$$\leq m \max_r \left| \left\langle \mathbf{w}_{y,r}^{(t)}, \boldsymbol{\xi} \right\rangle \right|^q$$

$$\leq m \widetilde{O}(\sigma_0^q \sigma_n^q d^{q/2} + n^q (pd)^{-q/2})$$

$$\leq 1,$$

where the last inequality follows because $\sigma_0 \leq \widetilde{O}(m^{-1/q} \mu^{-1})$ and $d \geq \widetilde{\Omega}(m^{2/q} n^2)$. Thus, with probability at least $1 - 1/d$,

$$\ell(F(\widetilde{\mathbf{W}}^{(t)}; \mathbf{x})) \geq \log(1 + (K-1)e^{-1}).$$

$\square$

