# OpenReview forum: "Theoretical Characterization of How Neural Network Pruning Affects its Generalization"
_ICLR.cc/2023/Conference — Submitted to ICLR 2023_

### Official Review · Reviewer_Rwrk · 2022-10-17

**Confidence:** 4
**Correctness:** 4
**Technical Novelty And Significance:** 3
**Empirical Novelty And Significance:** 3
**Recommendation:** 6

**Clarity, Quality, Novelty And Reproducibility:**

Clarity: Good. It is a pretty technique paper, but the writing is overall fairly good. I enjoyed the reading.

Quality: Good. I did not check all the proofs in the appendix, but I think this is a rigorous paper.

Novelty: Good. I think theory analysis for pruning is interesting and important.

Reproducibility: Not applicable. This is a theory paper, and all the proofs are included in the appendix.

**Strength And Weaknesses:**

Strength:
This is a well organized and well written paper. I did not have enough time to read all the details in the appendix, but I did go through the theorem statement in the main paper, which looks reasonable to me.
Pruning is pretty effective empirically, but as the authors mentioned in Section 3.2, if one only applies the "NTK analysis", it seems that pruning will only hurt. The nice thing about this paper is it uses the new technique called feature learning, and based on this technique, they can indeed show the effectiveness of pruning.
Empirically, feature learning is a much more realistic setting, so I think such analysis is interesting, and can be seen as a nice theoretical support for pruning methods.

Weakness:
The weakness of this paper, as well as many neural network theory papers, is the strong assumptions on the model. The assumptions on the input data and for the network structure look especially strange to me. However, given this is the first analysis for pruning, I guess that might be OK. It can be the first starting point for analyzing pruning methods.

**Summary Of The Paper:**

This paper analyzes pruning for overparameterized two-layer neural networks. The setting is "pruning the second last layer" of a deep neural network, so they assume the input has the form of (x1, x2) \in R^{2d}, where x1 and x2 are either ue_i, or a random vector. In other words, the input has a specical form of "half one-hot, half random noise" form. Moreover, they assume the network structure has the form sigma(<w, x1>)+sigma(<w, x2>), which means the weight w is applied to both x1 and x2 simultaneously. There are a bunch of other assumptions described in condition 2.2, including number of classes, training samples, etc.

Under these assumptions, the authors were able to prove interesting results on pruning the network. Specifically, they show that, after mild pruning, the generalization performance is increasing; after over pruning, the generalization performance is not better than random guess, but the training loss goes to 0. The nice thing about their analysis is, it is based on feature learning, instead of NTK analysis. In other words, they show the network learns the feature vector after mild pruning, and the network fails to learn the feature vector after over pruning.


**Summary Of The Review:**

This is an interesting theory paper of pruning for overparameterized two layer networks, with special assumptions on the input and network structure. The analysis is based on feature learning, which says with mild pruning, the network will still learn the feature signal but the noise signal will be reduced; with over pruning, the network is essentially learning the noise.

---

> ### Author Response · Authors · 2022-11-16
> **Response to Reviewer Rwrk**
>
> We thank Reviewer Rwrk for the positive review and we very appreciate the support of our work and recognition of our contribution!

---

### Official Review · Reviewer_bjfM · 2022-10-25

**Confidence:** 3
**Correctness:** 2
**Technical Novelty And Significance:** 3
**Empirical Novelty And Significance:** 3
**Recommendation:** 3

**Clarity, Quality, Novelty And Reproducibility:**

The paper provides generalization bound on random pruning for a specific model with some assumption on weight initialization. I understand these assumptions are required to provide such a theoretical bound but it is not clear how it can be extended in the existing model.


**Strength And Weaknesses:**

Strength:
The paper considered a classification task for overparameterized two-layer neural networks, where the network is randomly pruned according to different rates at the initialization. In this specific scenario, they theoretically prove that there exists a threshold on the pruning fraction such that pruning helps the neural network’s generalization. the generalization bound gets better as the pruning fraction gets larger.  Also for the overpruning there exists a relatively large pruning fraction such that the learned model yields poor generalization


Weakness:
1- Even though the theoretical result of the paper sounds interesting, they considered a very specific scenario and it is not clear this can be generalized to more complex models with different weight distributions.

2-The experimental results are very limited and they do not fully support the theoretical results. It is not clear how the results in figure 2 mapped to the bound proposed in theory. The paper should show the simulation results vs the theoretical result to make it more clear.


3- Also the experimental result of the real dataset is not clear. How come the accuracy in y axis is 100%? They only show as the pruning ratio increases the accuracy drops. How do these figures show the claims and arguments made by theory?


**Summary Of The Paper:**

The paper considered a classification task for overparameterized two-layer neural networks, where the network is randomly pruned according to different rates at the initialization and studied how different pruning fractions affect the model’s gradient descent dynamics and generalization.


**Summary Of The Review:**

The paper theoretically showed that there exists a threshold on the pruning fraction such that pruning helps the neural network’s generalization and also there exists a relatively large pruning fraction such that the learned model yields poor generalization. Even though these results sound interesting it is not clear how their experimental results support such a theory.

---

> ### Author Response · Authors · 2022-11-16
> **Response to Reviewer bjfM**
>
> We thank Reviewer bjfM for the review. We address the reviewer's concerns here.
>
> **Restrictive data and network assumption.** Please see our general response.
>
> **Q** It is not clear how the results in figure 2 mapped to the bound proposed in theory.
>
> **A**
> The three plots in Figure 1 are to illustrate Theorem 3.1 and Theorem 4.1.
> In particular, the blue curves in (a) and (b) represents the test error and shows a downward trend at the beginning of the pruning: (0, 0.3) for plot (a) and (0,0.4) for plot (b).
> This shows that mild pruning can indeed improve generalization as it is stated in Theorem 3.1. We point out that the bound in theoretical work usually carries arbitrary constant and thus the exact numerical value may not match the theoretical bound exactly.
> Also, in both plots, the training error is zero (plot (b) omitted training error plot).
> At pruning rate 0.9, both plots show much worse generalization than unpruned model while the training error effectively stays at zero which illustrate the over pruning phenomenon in Theorem 4.1.
>
> **Q** How come the accuracy in y axis is 100\%?
>
> **A**
> The lines with 100\% accuracy is the training accuracy.
> For overparameterized neural network, it is common to achieve zero training error (as it is proven in our theory that the training loss can be arbitrarily small).
> The purpose of the real world experiment is to show that mild pruning and over pruning can happen in practice.
> The extensive experiments in (Jin, et al. 2022) further confirmed the mild pruning phenomenon.
>
> Jin, Tian, et al. "Pruning's Effect on Generalization Through the Lens of Training and Regularization." arXiv preprint arXiv:2210.13738 (2022).

---

### Official Review · Reviewer_qT6W · 2022-11-03

**Confidence:** 3
**Clarity, Quality, Novelty And Reproducibility:** The paper is mostly clear and the res…
**Correctness:** 2
**Technical Novelty And Significance:** 3
**Empirical Novelty And Significance:** Not applicable
**Recommendation:** 5

**Strength And Weaknesses:**

Strength: The paper is mostly clear and the results are novel. The analysis of the convergence and generalization of training a pruned-at-initialization network is interesting and new. The analysis studies the neural network training under the so-called "active regime", which is more practically relevant.

Weakness: I think the theoretical results in the paper are great but I have some concerns about the way this paper is presenting it:

The assumption on the data distribution and the network structure is fairly restrictive. This paper studies the two-layer convolutional network but requires the size of the convolutional kernel to be exactly half of the input dimension and so is the stride so that the convolution layer outputs a 2d binary vector, and the two binary outputs can be viewed as the binary predictions of a linear classifier on the first half and the second of the data vector. Then the data distribution is carefully designed so that each data vector has either the first half as pure signal and the second half as noise or the other way around. Lastly, the weight for the output layer is sparse, binary, and fixed during training.

I understand many of these assumptions are useful for the analysis (and I really think they are cleverly made) but most of them are impractical, making the theoretical results less relevant in interpreting the performance of a practical neural network trained under a real dataset, and this is shown in the numerical section that experiments on the real dataset cannot be well explained by the theorems in this paper.

Therefore, I think a more detailed explanation of why some assumptions are made about the data and network, together with remarks on how the theoretical results can be (or can be made in future work) relevant to practice is necessary for this paper, otherwise, the theoretical results can be misleading.

**Summary Of The Paper:**

This paper studies the effect of random pruning at initialization on the generalization of a neural network. The author theoretically shows that pruning of some two-layer convolutional neural networks under small initialization could improve their generalization if the probability of dropping a neural is mildly large, and could deteriorate the generalization if it is too large.

**Summary Of The Review:**

Strength: well-written, results are interesting and new;
Weakness: Restrictive assumptions without justification on their practical relevance.

---

> ### Author Response · Authors · 2022-11-16
> **Response to Reviewer qT6W**
>
> We thank Reviewer qT6W for the review. We address the reviewer's concerns here.
>
> **Unrealistic data and network assumption.** Please see our general response.
>
>
> **Q**
> The real dataset cannot be well explained by the theorems in this paper.
>
> **A**
> The purpose of the real world experiment is to show that mild pruning and over pruning can happen in practice.
> Although the generalization improvement is not obvious in Figure 3, both plots show that there exists a range of pruning ratio such that the network achieves nearly the same test performance as the dense model.
> For plot (a), the range is $[0, 79]$ and for plot (b), the range is $[0, 36]$. Further, the extensive experiments in (Jin et al. 2022) further confirmed that the mild pruning phenomenon can happen for real world datasets.
>
> Jin, Tian, et al. "Pruning's Effect on Generalization Through the Lens of Training and Regularization." arXiv preprint arXiv:2210.13738 (2022).

---

### Official Review · Reviewer_oTTX · 2022-11-04

**Confidence:** 4
**Clarity, Quality, Novelty And Reproducibility:** Overall, the paper is well-written an…
**Correctness:** 3
**Technical Novelty And Significance:** 2
**Empirical Novelty And Significance:** 2
**Recommendation:** 5

**Strength And Weaknesses:**

This is the first work towards understanding the generalization of pruned neural networks trained by gradient descent. Their results are interesting: there is a phase transition of pruning ratio. When pruning ratio is lower than some threshold, it helps generalization; while if the ratio is higher than some threshold, it will hurt generalization, which matches our intuition. Their proof is based on assuming sparse data with Gaussian noise, and controlling the correlations between Gaussian random initialized weights and noise. A small pruning ratio provides some regularization-like benefits by reducing the variance of the noise, while the more aggressive pruning will limit the prediction of the network below 1, i.e., $F(\tilde{\mathbf{W}},\mathbf{x}) < 1$, so the generalization will never be smaller than a threshold. The proof idea is interesting to me, and the results are also novel.

Some issues:

1. The assumption on input data is way strong: the authors assume the input data is composed by basis vector concatenated with Gaussian noise, i.e., $\mathbf{x} = [\mathbf{e}_i, \boldsymbol{\xi}]$. I guess this data assumption follows recent work:
\emph{ Cao, Yuan, et al. "Benign Overfitting in Two-layer Convolutional Neural Networks." arXiv preprint arXiv:2202.06526 (2022).} However, Cao et al do not assume the deterministic part of the data is basis vector.

2. The neural network structure is bit toy. Usually, for two layer ReLU network, we assume $f(\mathbf{x}) = \mathbf{a}^T \sigma (\mathbf{W}\mathbf{x})$, while in this paper's setting, the top layer is removed. Hence I cannot even call this model a ``two layer neural network''. Perhaps linear model with rectified output is more accurate. It is also surprising that here we only achieve $O(1/\epsilon)$ convergence of training loss, while it is well-known that even for deep neural network, the convergence rate is $O(\log (1/\epsilon))$ in overparameterized setting, or even under only mild overparameterization regime:

\emph{ Nguyen, Quynh. "On the proof of global convergence of gradient descent for deep relu networks with linear widths." International Conference on Machine Learning. PMLR, 2021.}


3. Technical question: in Lemma D.25, the key Lemma to prove generalization, the authors said that the correlation between output model and Gaussian noise: $\langle \tilde{\mathbf{w}}^*, \boldsymbol{\xi} \rangle$ satisfy Gaussian distribution. If $\tilde{\mathbf{w}}^*$ is a constant, it definitely holds. However, if my understanding is correct, output weight $\tilde{\mathbf{w}}^*$ is also a random variable, and the product of two Gaussian RVs is not necessarily Gaussian. So I am curious how the authors eliminate the randomness of $\tilde{\mathbf{w}}^*$? Or do we need to take expectation over $\tilde{\mathbf{w}}^*$ to make it constant?


4. Perhaps not extremely relevant but I think the following paper about Dropout should be cited:
Arora, Raman, et al. "Dropout: Explicit forms and capacity control." International Conference on Machine Learning. PMLR, 2021.


**Summary Of The Paper:**

 This paper studies the key problem in deep learning: why pruning can help generalization. They consider a one-layer (I will explain why I call it one layer) neural network with ReLU activation, and assume certain structure of input data and labels. They model the pruning operator as the element wised multiplication between weight matrices and masking matrices. They show that, when pruning ratio is lower than some threshold, it helps generalization; while if the ratio is higher than some threshold, it will hurt generalization. Their proof technique is to study the correlations between model weights and input data noise. If the pruning ratio is lower than some threshold, the pruning will reduce the variance of the noise and hence improve the generalization. When the ratio is higher than some threshold, it will constrain the model capacity such that the prediction will be bounded (less than 1), and hence the generalization is lower bounded by a constant as well.


**Summary Of The Review:**

In general, this is an interesting paper towards understanding how GD+pruning helps generalization, and it could inspire a line of follow-up works. However, as I mentioned in weakness part, the setup is way toy, even among the theory papers.

---

> ### Author Response · Authors · 2022-11-16
> **Response to Reviewer oTTX**
>
> We thank reviewer oTTX for the careful review. We address the reviewer's concerns below.
>
> **Restrictive data and network assumption.** Please see our general response.
>
> **Only able to achieve $O(1/\epsilon)$ training loss convergence rate instead of $O(\log (1/\epsilon))$ convergence rate.**
> This is because our setting considers binary classification with **cross-entropy loss** while (Nguyen 2021) considers the square loss. The square loss is both smooth and strongly convex and thus $O(\log 1/\epsilon)$ convergence is possible when the network is sufficiently overparameterized to be put onto the NTK regime.
> The cross-entropy loss is only smooth but not strongly-convex and thus $O(1/\epsilon)$ convergence is standard.
>
> **$\langle{\tilde{\mathbf{w}}^\star, \boldsymbol{\xi}} \rangle $ satisfies Gaussian distribution.**
> We mean conditioned on $\tilde{\mathbf{w}}^\star$ and only consider the randomness of $\boldsymbol{\xi}$, the inner product is Gaussian.
>
> Nguyen, Quynh. "On the proof of global convergence of gradient descent for deep relu networks with linear widths." International Conference on Machine Learning. PMLR, 2021.

---

### Author Response · Authors · 2022-11-16
**General Response**

We thank all the reviewers for their review comments. We address one of the most common concern here.

**The assumption on data and model is restrictive.**
Many thanks for this concern.
Given the tools and techniques currently available in deep learning theory, our model is simplified, yet still sensible to capture the effect of pruning on neural network's generalization. Our signal-noise data assumption is common and has been widely accepted in the theory community. The following is an incomplete list:

- Cao, Yuan, et al. "Benign Overfitting in Two-layer Convolutional Neural Networks." NeurIPS 2022, oral presentation.

- Frei, Spencer, Niladri S. Chatterji, and Peter Bartlett. "Benign overfitting without linearity: Neural network classifiers trained by gradient descent for noisy linear data." Conference on Learning Theory. PMLR, 2022.

- Allen-Zhu, Zeyuan, and Yuanzhi Li. "Towards understanding ensemble, knowledge distillation and self-distillation in deep learning." arXiv preprint arXiv:2012.09816 (2020).

- Jelassi, Samy, and Yuanzhi Li. "Towards understanding how momentum improves generalization in deep learning." International Conference on Machine Learning. PMLR, 2022.

The one-hidden-layer neural network architecture in our work is also common in the above references as well as the following literature.

- Huang Y, Lin J, Zhou C, et al. Modality Competition: What Makes Joint Training of Multi-modal Network Fail in Deep Learning?(Provably)[J]. arXiv preprint arXiv:2203.12221, 2022.
- Zou, Difan, et al. "Understanding the Generalization of Adam in Learning Neural Networks with Proper Regularization." (2021).
- Chen, Zixiang, et al. "Towards Understanding Mixture of Experts in Deep Learning." arXiv preprint arXiv:2208.02813 (2022).

Our motivation is as follows.
It has been observed as an interesting phenomenon in practice that mild pruning can slightly increase the performance of the trained model.
However, it is unclear whether such an improvement is actually due to pruning or it is only pure experimental variation.
*The goal of our work is to affirmatively explain such a phenomenon theoretically as the benefit of pruning. Specifically, we show that the mild pruning effect can provably happen for overparameterized neural networks and understand why and how mild pruning can improve generalization.*
To achieve this goal, we study pruning using the signal-noise data model.
In particular, we find out that when the signal vector is 1-sparse, mild pruning can indeed happen.
Note that the location of the non-zero entry is not important and our analysis holds for all 1-sparse vector.
Our analysis can be generalized to $k$-sparse vector to make it more realistic and this will make the final results have dependence on $k$.
Note that this is a realistic assumption since it is known that real world datasets are often sparse after some appropriate transform.
In addition to our novel results, *we also want to highlight that our work also provide new insights on why and how pruning can improve generalization*: our study finds out that mild pruning can improve generalization by reducing the noise in the learned weights while relatively preserving the strength of the signal.
We believe such understanding is also new.
We would like to point out that a recent empirical work (Jin et al. 2022) (which came after our work) has shown that the mild pruning effect can happen across many different data sets in practice.
We believe our theory has its own real world implication and merit.

Jin, Tian, et al. "Pruning's Effect on Generalization Through the Lens of Training and Regularization." arXiv preprint arXiv:2210.13738 (2022).

---

### Decision · Program_Chairs · 2023-01-20

**Decision:**

Reject

**Justification For Why Not Higher Score:**

The paper can not be accepted because it does not meet the standards of quality, novelty, and significance for ICLR. The paper does not convincingly demonstrate the validity and relevance of the results, and it does not provide enough empirical and theoretical support for the claims. The paper also does not engage with the existing literature and the broader implications of the work. The paper needs to address the reviewers' concerns and feedback, and to provide more rigorous and comprehensive analysis and experiments. The paper also needs to justify and motivate the assumptions and the setting, and to discuss the limitations and extensions of the work. The paper has the potential to be an interesting and important contribution, but it requires more work and refinement.

**Justification For Why Not Lower Score:**

N/A

**Metareview: Summary, Strengths And Weaknesses:**

The paper studies the effect of random pruning at initialization on the generalization of a two-layer neural network. The paper claims to show that there is a phase transition of pruning ratio, such that mild pruning improves generalization while over pruning hurts it. The paper uses a feature learning approach to analyze the convergence and generalization of the pruned network.

The reviewers appreciate the novelty and interest of the topic, and the clarity and rigor of the writing and the proofs. However, they also raise several major concerns that prevent the paper from being accepted:

- The paper makes very strong and unrealistic assumptions on the data distribution and the network structure, which limit the applicability and relevance of the results. The data is assumed to be composed of half one-hot and half random noise vectors, and the network is assumed to apply the same weight to both halves of the input. These assumptions are not justified or motivated by any practical scenario, and they are not discussed in relation to existing work or real data. The paper should provide more explanation and evidence for why these assumptions are necessary and reasonable, and how they can be relaxed or generalized in future work.

- The paper does not provide sufficient empirical evidence to support the theoretical claims. The experiments on synthetic data are very limited and do not show how the theoretical bounds compare to the actual performance. The experiments on real data are unclear and do not demonstrate the phase transition of pruning ratio. The paper should show more comprehensive and convincing experiments that illustrate the main results and compare them with existing methods or baselines.

- The paper does not discuss the related work and the implications of the results in a broader context. The paper should cite and compare with other works on pruning, feature learning, and generalization, and discuss the limitations and extensions of the results. The paper should also explain how the results can inform the design and understanding of pruning methods for more complex and realistic models and data.

In summary, the paper has some interesting and novel aspects, but it also has many flaws and gaps that need to be addressed. The paper is not ready for publication in its current form, and it requires major revisions and improvements. Therefore, the paper is rejected.